# Molecular basis for the role of disulfide-linked αCTs in the activation of insulin-like growth factor 1 receptor and insulin receptor

Jie Li[1], Jiayi Wu[2], Catherine Hall[2], Xiao-chen Bai[1,3]*, Eunhee Choi[2]*

[1]Department of Biophysics, University of Texas Southwestern Medical Center, Dallas, United States; [2]Department of Pathology and Cell Biology, Vagelos College of Physicians and Surgeons, Columbia University, New York, United States; [3]Department of Cell Biology, University of Texas Southwestern Medical Center, Dallas, United States

*For correspondence:
Xiaochen.Bai@UTSouthwestern.edu (X-chenB);
ec3477@cumc.columbia.edu (EC)

Competing interest: The authors declare that no competing interests exist.

**Abstract** The insulin receptor (IR) and insulin-like growth factor 1 receptor (IGF1R) control metabolic homeostasis and cell growth and proliferation. The IR and IGF1R form similar disulfide bonds linked homodimers in the apo-state; however, their ligand binding properties and the structures in the active state differ substantially. It has been proposed that the disulfide-linked C-terminal segment of α-chain (αCTs) of the IR and IGF1R control the cooperativity of ligand binding and regulate the receptor activation. Nevertheless, the molecular basis for the roles of disulfide-linked αCTs in IR and IGF1R activation are still unclear. Here, we report the cryo-EM structures of full-length mouse IGF1R/IGF1 and IR/insulin complexes with modified αCTs that have increased flexibility. Unlike the Γ-shaped asymmetric IGF1R dimer with a single IGF1 bound, the IGF1R with the enhanced flexibility of αCTs can form a **T**-shaped symmetric dimer with two IGF1s bound. Meanwhile, the IR with non-covalently linked αCTs predominantly adopts an asymmetric conformation with four insulins bound, which is distinct from the **T**-shaped symmetric IR. Using cell-based experiments, we further showed that both IGF1R and IR with the modified αCTs cannot activate the downstream signaling potently. Collectively, our studies demonstrate that the certain structural rigidity of disulfide-linked αCTs is critical for optimal IR and IGF1R signaling activation.

## Editor's evaluation

This paper investigated the structural changes in the insulin and IGF1 receptors caused by ligand binding using cryo-electron microscopy and found that the disulfide-linked C-terminal segment of α-chains of these receptors are important for receptor activation. This is an important manuscript that addresses a research question of interest to the fields of metabolism, cancer, growth and aging. It provides convincing mechanistic insights into the roles of the disulfide linked C-terminal segment of the α-chains of the IR and IGF1R and their activation.

## Introduction

The insulin receptor (IR) and insulin-like growth factor 1 receptor (IGF1R) are a subfamily of receptor tyrosine kinases (RTKs) (*Ullrich et al., 1985*; *Shier and Watt, 1989*; *Abbott et al., 1992*; *Jui et al., 1994*; *Ebina et al., 1985*; *Ullrich et al., 1986*; *Cai et al., 2022*) and have pivotal roles in physiology (*Accili et al., 1996*; *Haeusler et al., 2018*; *Olefsky, 1976*; *Pollak, 2012*; *Joshi et al., 1996*; *Boucher*

*et al., 2014*). Dysfunction of the IR family signaling causes human diseases such as cancer, diabetes, and aging (*Boucher et al., 2014*; *Petersen and Shulman, 2018*; *Cohen and LeRoith, 2012*; *Anisimov and Bartke, 2013*; *Kenyon, 2011*). Unlike most of other RTKs that are monomeric in the unliganded state, the IR family receptors form a stable dimer by inter-protomer interactions and disulfide bonds, independent of ligand binding (*Schäffer and Ljungqvist, 1992*; *Sparrow et al., 1997*; *Massagué and Czech, 1982*; *Chiacchia, 1991*; *Cheatham and Kahn, 1992*; *Finn et al., 1990*; *Hubbard, 2013*). Insulin or IGF1 binding induces a large structural change and stabilizes the active conformations of IR and IGF1R, which activate signaling cascades (*Uchikawa et al., 2019*; *Gutmann et al., 2020*; *Li et al., 2022*; *Li et al., 2019*; *Xu et al., 2018*; *Kavran et al., 2014*; *McKern et al., 2006*; *Weis et al., 2018*; *Zhang et al., 2020*; *Xu et al., 2020*; *Croll et al., 2016*).

Despite the high structural similarity among the IR family receptors (*Figure 1—figure supplement 1*), ligand binding properties and the ligand-mediated activation mechanisms of IR and IGF1R are significantly different. The IR has two distinct insulin binding sites — site-1 and site-2 (*Uchikawa et al., 2019*; *Gutmann et al., 2020*; *Li et al., 2022*). Maximally four insulins bind to the IR dimer at two distinct sites, promoting a fully activated, **T**-shaped symmetric IR. At subsaturated insulin concentrations, however, IR predominantly adopts asymmetric conformations with one or two insulins bound (*Li et al., 2022*; *Weis et al., 2018*; *Scapin et al., 2018*; *Nielsen et al., 2022*). The IR with one insulin bound to the site-1 shows a Γ-shaped dimer; while the IR with two insulins bound assumes a **T**-shaped dimer, wherein one insulin bound to the site-1 and the other insulin bound to the hybrid site formed by sites-1 and –2′ from two adjacent protomers (denoted as hybrid site hereafter) (*Li et al., 2022*). By contrast, even at saturated IGF1 concentrations, IGF1R exclusively forms a Γ-shaped asymmetric dimer with a single IGF1 bound to the site-1. IGF1 is undetectable on the side surface of FnIII-1 in IGF1R, which is equivalent to the site-2 of IR (*Li et al., 2019*).

A structural feature unique to IR family receptors is a motif localized in the C-terminal region of α-subunit, named αCT. The αCT (human IR, residues 693–710; human IGF1R, residues 682–704) form an α-helix and is constitutively associated with the N-terminal L1 domain of IR or IGF1R, together serving as the site-1 for ligand binding (*Uchikawa et al., 2019*; *Li et al., 2022*; *Li et al., 2019*; *Xu et al., 2018*; *Weis et al., 2018*; *Scapin et al., 2018*; *Menting et al., 2013*; *Williams et al., 1995*; *Mynarcik et al., 1996*; *Whittaker et al., 2001*; *Whittaker et al., 2008*; *Smith et al., 2010*). Moreover, cysteine triplet (human IR, Cys682/Cys683/Cys685; human IGF1R, Cys669/Cys670/Cys672) which lies in the N-terminal region of the αCT helix forms disulfide bonds that covalently link two protomers (*Figure 1—figure supplement 2*). Given this structural particularity, the disulfide-linked αCTs are likely to play an important role in regulating the activation and function of IR family receptors. Indeed, it has been shown that, in the active structure of IGF1R bound with IGF1, the two disulfide-linked αCTs bear a rigid and elongated conformation (*Li et al., 2019*). The physically coupled αCTs in the active IGF1R restrict the structural flexibility of the unliganded L1′/αCT that is required for the binding of another IGF1, resulting in the negative cooperativity in the binding of IGF1 to IGF1R. However, the lack of structural information of IGF1R in the absence of the disulfide linkage of αCTs has hindered the complete understanding of the specific role of disulfide-linked αCTs in the activation of IGF1R.

The two insulins bound IR is the major conformation at subsaturated insulin concentrations (*Li et al., 2022*). Such structural observation hints that, rather than conferring the negative cooperativity, the disulfide-linked αCTs might play a distinct role in the activation of IR. Intriguingly, in the asymmetric IR with two insulins bound, the disulfide-linked αCTs adopt a rigid and stretched conformation (*Li et al., 2022*), unique to that shown in one IGF1 bound IGF1R. We have proposed that the binding of another insulin to the hybrid site would require the conformational change of IR from asymmetric to symmetric, in order to prevent the overstretching of the disulfide-linked αCTs. However, this idea has not been tested experimentally.

Here, we first determined the cryo-EM structure of IGF1 bound full-length IGF1R mutant (four glycine residues inserted in the αCT; IGF1R-P673G4) that has been shown to exhibit reduced negative cooperativity (*Li et al., 2022*). Strikingly, a portion of the IGF1R-P673G4/IGF1 particles adopts a **T**-shaped symmetric conformation with two IGF1s bound at site-1s. Our functional results further showed that IGF1R with reduced negative cooperativity cannot achieve full activation in response to IGF1. This result confirms that the disulfide-linked αCTs are the origin of the negative cooperativity in IGF1R and demonstrates the critical role of the negative cooperativity in the activation of IGF1R.

We next determined the cryo-EM structure of insulin bound full-length IR mutant with physically decoupled αCTs (C684S/C685S/C687S; IR-3CS). The IR with non-covalently linked αCTs predominantly adopts asymmetric conformations even at saturated insulin concentrations. In the middle part of this asymmetric dimer, two insulins are simultaneously bound at sites-1 and –2' of the hybrid site. This structure is largely different from that of IR-wild type (WT) at saturated insulin concentrations, which displays exclusive **T**-shaped symmetric conformation. Altogether, we propose that the disulfide-linked αCTs of IR facilitate the structural transition from asymmetric to symmetric conformation upon insulin stimulation.

## Results

### The disulfide-linked αCTs are required for the formation of Γ-shaped IGF1R dimer

Our previous study has suggested the disulfide-linked αCTs as the origin of negative cooperativity in the binding of IGF1 to IGF1R (*Li et al., 2019*). Through the mutagenesis and binding experiments, we also showed that increasing the flexibility of disulfide-linked αCTs by inserting four glycine residues (IGF1R-P673G4) reduces the negative cooperativity of IGF1R that allows the binding of the second IGF1 to IGF1R. To further investigate the functional role of the disulfide-linked αCTs in the activation of IGF1R, we first expressed and purified a full-length IGF1R-P673G4, and determined the cryo-EM structure of IGF1 bound IGF1R-P673G4 (*Figure 1A and B*; *Figure 1—figure supplements 2 and 3*; *Figure 1—figure supplement 2—source data 1*). We will use the amino acid numbering of human IGF1R and IR without signal peptide to describe our structural and functional analysis. The 3D classification result of IGF1R-P673G4/IGF1 dataset showed that most particles (89%) assumed asymmetric conformations. In the top region of the asymmetric dimer, one of the two L1 domains with IGF1 bound was well-resolved in the cryo-EM map; while another L1 domain, localized in the middle part of the dimer, was only partially resolved, suggesting its dynamic nature (*Figure 1D*). This is in sharp contrast to the previous image processing result of IGF1R-WT/IGF1 data showing that the unliganded L1 domain in the Γ-shaped dimer bears a rigid conformation and is sandwiched tightly between the two membrane-proximal domains of IGF1R (*Figure 1C*; *Li et al., 2019*).

In addition, the 3D reconstruction of the particles from the minor class of IGF1R-P673G4/IGF1 dataset resulted in a **T**-shaped symmetric conformation, which is resemblance of the **T**-shaped IR with two insulins bound in the top part (*Figures 1D, 2A and B*; *Li et al., 2022*). Notably, such conformation does not exist in the dataset of IGF1R-WT/IGF1 complex (*Li et al., 2019*). The subsequent 3D refinement of this minor class resolved a cryo-EM map at overall ~4 Å resolution (*Figure 2A and B*). In this **T**-shaped IGF1R-P673G4/IGF1 complex, two IGF1s are bound in the top part of the 'T', in a similar fashion as the binding of single IGF1 to IGF1R-WT. The structural differences between IGF1R-WT/IGF1 and IGF1R-P673G4/IGF1 suggest that the disulfide-linked αCTs is the origin of negative cooperativity in the binding of IGF1 to IGF1R, and is critical for stabilizing the one IGF1 bound Γ-shaped asymmetric conformation of IGF1R.

### The disulfide-linked αCTs are required for optimal IGF1R signaling and trafficking

To validate the functional importance of the disulfide-linked αCTs for IGF1R activation, we analyzed IGF1-induced IGF1R autophosphorylation (pY1135/1136, pY IGF1R) and the activating phosphorylation of a potent downstream substrate of IGF1R, extracellular signal-regulated kinase 1/2 (pERK) in 293 FT cells expressing IGF1R-WT or mutants at multiple time points (*Figure 3A and B*; *Figure 3— source data 1 and 2*). In addition to IGF1R-P673G4, we examined IGF1R-Δ3C, a mutant designed to have flexible αCTs by removing three cysteines in the αCT (*Figure 1A and B*). Previous study demonstrated that deletion of these cysteines increased the binding of IGF1 to IGF1R and reduced the negative cooperativity (*Li et al., 2019*). The autophosphorylation of IGF1R-WT peaked at 60 min after IGF1 stimulation and did not fall below 80% of the peak level after 3 hr. The levels of autophosphorylation of IGF1R-P673G4 and IGF1R-Δ3C were greatly reduced at multiple time points after IGF1 stimulation (*Figure 3A and B*; *Figure 3—source data 1 and 2*). Consequently, the pERK level was also significantly decreased in cells expressing IGF1R-P673G4 and IGF1R-Δ3C. These functional data together

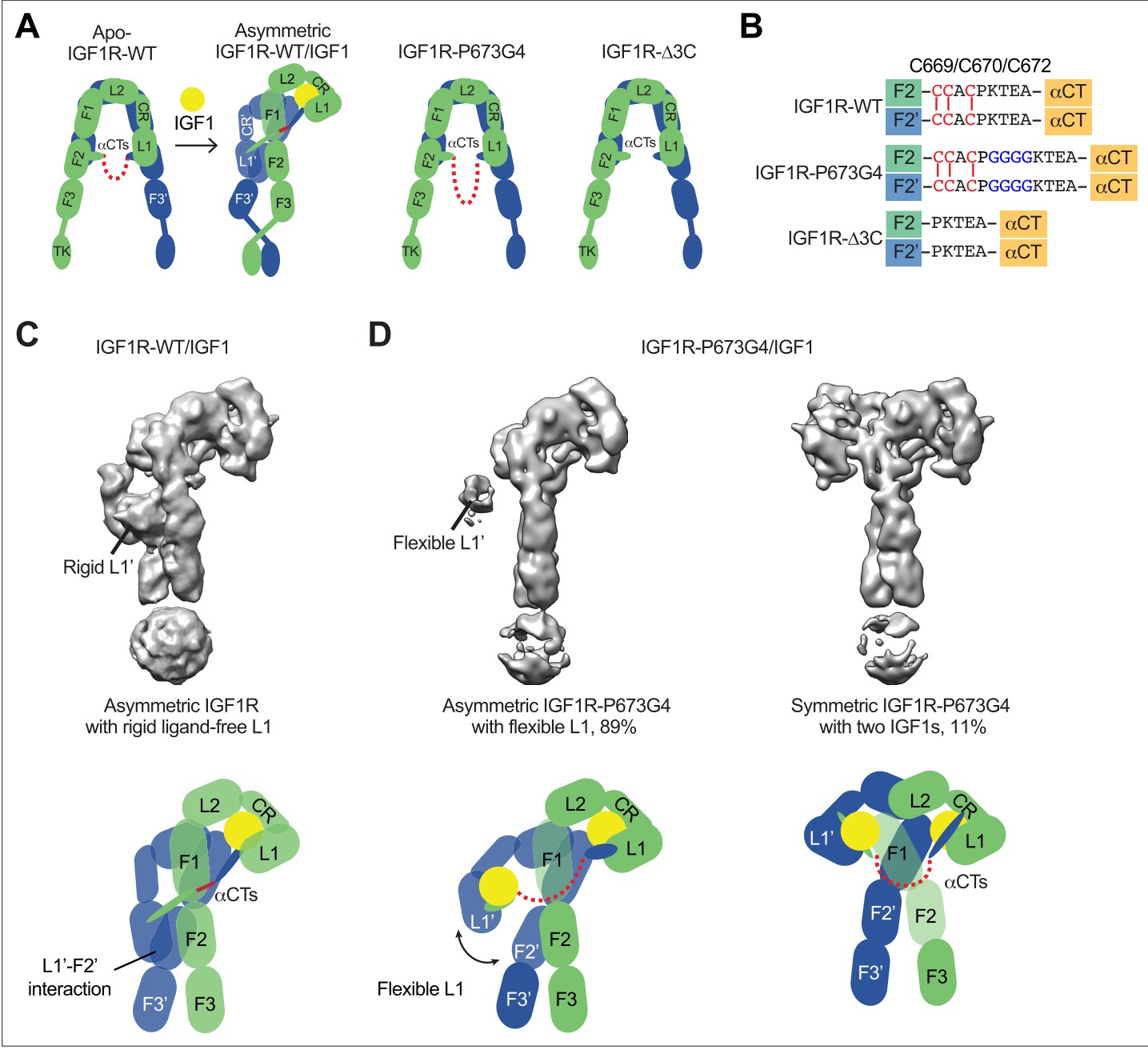

**Figure 1.** Structures of IGF1R-P673G4 with IGF1 bound. (**A**) Cartoon representation of the apo-IGF1R, IGF1R/IGF1 complex showing the IGF1R dimer bound with one IGF1 (left), apo-IGF1R-P673G4 (middle) and apo-IGF1R-Δ3C (right). The two IGF1R protomers are colored in green and blue; the IGF1 is colored in yellow. The name of domain is labeled as follows: Leucine-rich repeat domains 1 and 2 (L1 and L2); cysteine-rich domain (CR); Fibronectin type III-1,–2, and –3 domains (F1, F2, F3), tyrosine kinase (TK) domain, and the C-terminus of the α-subunit (αCT) (referred to the domains in protomers 1 and 2 as L1-TK, and L1'-TK', respectively). The linker between two αCTs is shown as red dotted line. (**B**) Sequence of the linker between two αCTs in the IGF1R-WT, IGF1R-P673G4, and IGF1R-Δ3C showing disulfide bonds in red. (**C**) Cryo-EM density map of the asymmetric IGF1R-WT/IGF1 complex. A cartoon model illustrating the structure (below). Rigid L1' (L1'-F2' interaction) is noted. (**D**) Cryo-EM density maps of the IGF1R-P673G4/IGF1 complex in both asymmetric (left) and symmetric (right) conformations. The asymmetric and symmetric conformations comprise of 89% and 11% of the good particles, respectively. Cartoon models illustrating the structural difference between symmetric and asymmetric conformations (below).

The online version of this article includes the following source data and figure supplement(s) for figure 1:

**Figure supplement 1.** Sequence alignment of human IGF1R and IR.

**Figure supplement 2.** Domains and protein preparation of IGF1R and IR.

**Figure supplement 2—source data 1.** Source data for *Figure 1—figure supplement 2B and D*.

**Figure supplement 3.** Cryo-EM analysis of the IGF1R-P673G4/IGF1 complex.

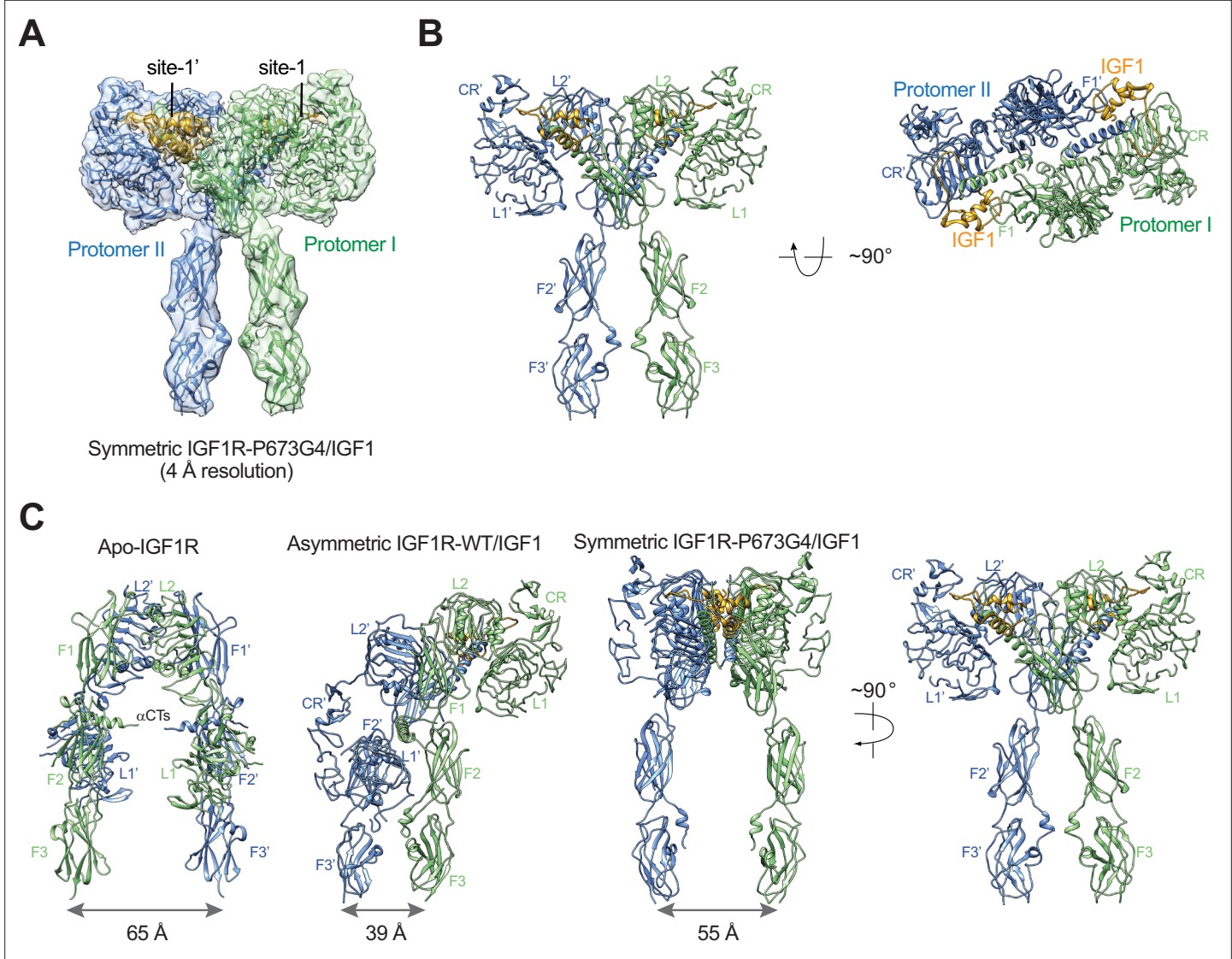

**Figure 2.** The **T**-shaped symmetric IGF1R-P673G4 dimer with two IGF1s bound. (**A**) A 3D reconstruction of the IGF1R-P673G4/IGF1 complex in symmetric conformation fitted into a cryo-EM map at 4 Å. (**B**) Ribbon representation of the symmetric IGF1R-P673G4/IGF1 complex, shown in two orthogonal views. (**C**) Structural comparisons between apo-IGF1R (PDB: 5U8R), asymmetric IGF1R-WT/IGF1 (PDB: 6PYH), and symmetric IGF1R-P673G4/IGF complexes.

with the structural results (*Figures 1 and 2*) suggest that disulfide-linked αCTs facilitates optimal IGF1R activation and signaling through stabilizing the one-ligand bound asymmetric conformation.

The activated IGF1R undergoes clathrin-mediated endocytosis that redistributes and terminates the IGF1R signaling (*Goh and Sorkin, 2013*). We generated HeLa cells stably expressing IGF1R-GFP-WT or -P673G4, and examined the subcellular localization of these IGF1R-GFP proteins (*Figure 3C and D*; *Figure 3—source data 2*). Without IGF1 stimulation, both IGF1R-WT and IGF1R-P673G4 are localized to the plasma membrane in a similar level. IGF1R-WT underwent IGF1-induced internalization and most IGF1R-WT signal was found inside cells after 1 hr. The IGF1R kinase inhibitor (BMS536924) prevented the IGF1-induced IGF1R endocytosis, confirming the requirement of kinase activation in promoting IGF1R endocytosis (*Figure 3D*; *Figure 3—source data 2*). Strikingly, the level of IGF1R-P673G4 on the plasma membrane was much stronger than those in IGF1R-WT at several time points after IGF1 stimulation, and the majority of IGF1R-P673G4 remained on the plasma membrane for 3 hr (*Figure 3C and D*; *Figure 3—source data 2*), suggesting defective endocytosis that probably results from impaired activation of the IGF1R kinase and its downstream signaling (*Yoneyama et al., 2018*;

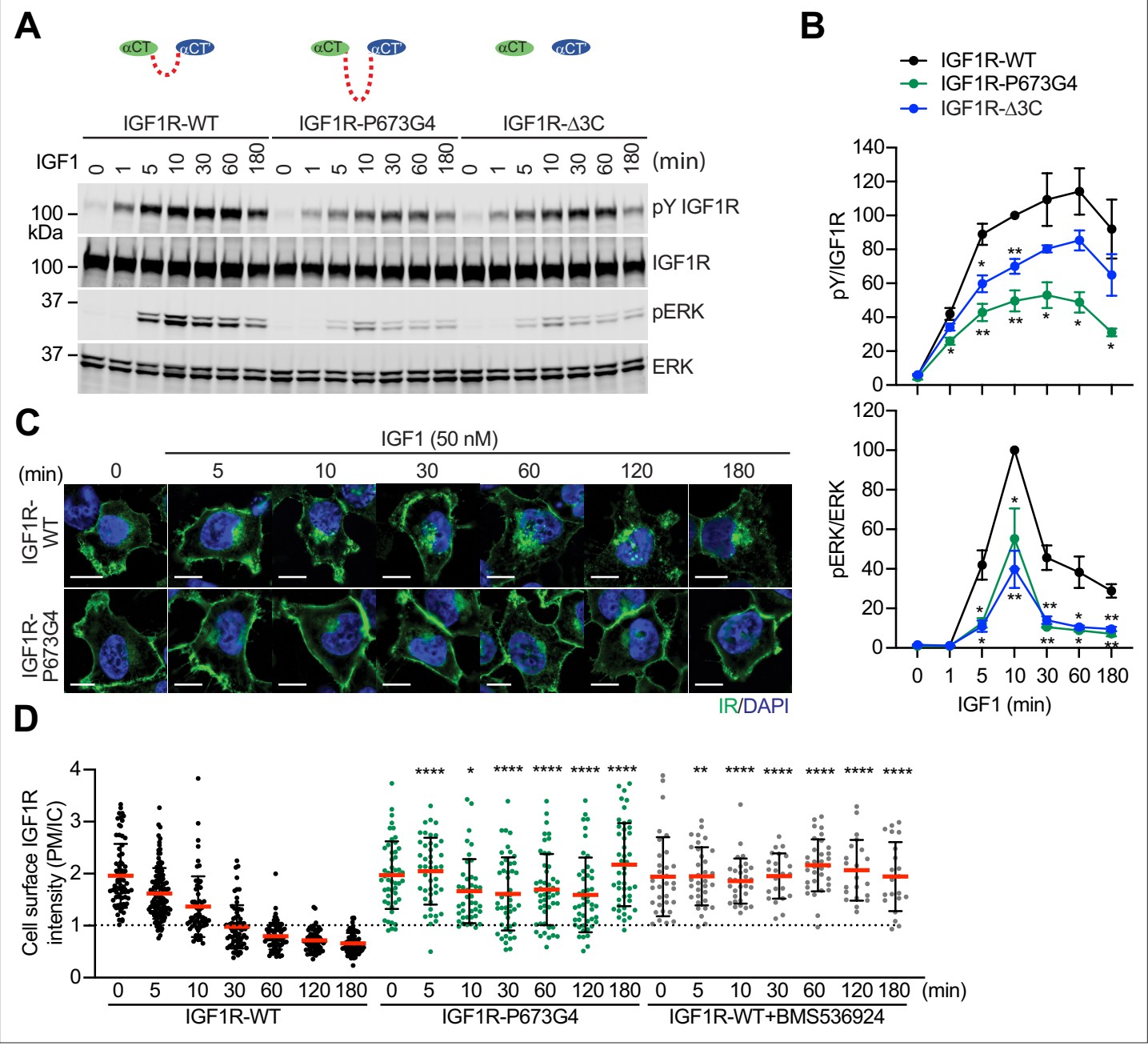

**Figure 3.** Functional importance of disulfide-linked αCTs on the IGF1R activation. (**A**) IGF1-induced IGF1R autophosphorylation and pERK levels in 293 FT cells expressing IGF1R wild-type (WT), IGF1R-P673G4, and IGF1R-Δ3C. Cells were treated with 50 nM IGF1 for the indicated times. Cell lysates were blotted with the indicated antibodies. Source data are presented in *Figure 3—source data 1*. (**B**) Quantification of the western blot data shown in **A** (Mean ± SEM, N=3). Significance calculated using two-tailed student t-test; *p<0.05 and **p<0.01. Source data are presented in *Figure 3—source data 2*. (**C**) HeLa cells expressing IGF1R-GFP WT or IGF1R-GFP P673G4 were starved, treated with 50 nM IGF1 for indicated times, and stained with anti-GFP (IGF1R, green) and DAPI (blue). 2 µM BMS536924 were treated for 2 hr prior to IGF1 stimulation. Scale bar, 10 µm. (**D**) Quantification of the ratios of plasma membrane (PM) and intracellular (IC) IGF1R-GFP signals of cells in **C** (IGF1R WT0, n=72; WT5, n=118; WT10, n=62; WT30, n=71; WT60, n=71; WT120, n=69; WT180, n=67; P673G4-0, n=50; P673G4-5, n=44; P673G4-10, n=42; P673G4-30, n=46; P673G4-60, n=48; P673G4-120, n=47; P673G4-180, n=45; WT +BMS0, n=32; WT +BMS5, n=33; WT +BMS10, n=31; WT +BMS30, n=23; WT +BMS60, n=33; WT +BMS120, n=21; WT +BMS180, n=21). Mean ± SD; two-tailed student t-test; *p<0.05; **p<0.01; ****p<0.0001. Source data are presented in *Figure 3—source data 2*.

The online version of this article includes the following source data for figure 3:

**Source data 1.** Source data for *Figure 3A*.

**Source data 2.** Source data for *Figure 3B and D*.

*Monami et al., 2008*; *Zheng et al., 2012*; *Cai et al., 2017*; *Sehat et al., 2007*). Taken together, these results together with our previous (*Li et al., 2019*) and current structural studies demonstrate that the disulfide-linked αCTs ensure the formation of a stable one-IGF1 bound Γ-shaped IGF1R dimer by conferring the negative cooperativity, leading to optimal IGF1R activation and trafficking.

## The disulfide-linked αCTs promote the formation of a T-shaped IR dimer

We and others have shown that IR with insulin mutant only bound at site-1 or at subsaturated insulin concentrations adopts an asymmetric conformation (*Li et al., 2022*; *Weis et al., 2018*; *Scapin et al., 2018*; *Nielsen et al., 2022*). In one-half of such asymmetric dimer, one insulin simultaneously contacts both site-1 and site-2' from adjacent protomers (*Li et al., 2022*). We also show that, in this asymmetric dimer, the disulfide-linked αCTs assume a remarkably stretched conformation, which restricts the structural rearrangement of the hybrid site that is required for the binding of another insulin. This explains why the asymmetric conformation cannot be formed at saturated insulin concentrations when four insulin molecules are bound at both sites-1 and –2. We hypothesize that if the two αCTs are physically decoupled by disrupting the disulfide bonds, the asymmetric conformation could be formed even at saturated insulin concentrations.

To test this hypothesis, we mutated all the three cysteine residues in the αCT of IR that form disulfide bonds (mouse IR-C684S/C685S/C687S; IR-3CS) and determined the cryo-EM structure of IR-3CS at saturated insulin concentrations (*Figure 4—figure supplement 1*). In contrast to the 3D classification result of fully liganded IR-WT that shows a single **T**-shaped conformation (*Uchikawa et al., 2019*; *Gutmann et al., 2020*; *Li et al., 2022*), the 3D classification of fully liganded IR-3CS revealed three distinct classes (*Figure 4A and B*). It is worth noting that, in all the cryo-EM maps determined from this dataset, the N-terminal region of the αCTs containing the mutated cysteines were completely unresolved, confirming that the two αCTs are physically disconnected. Two of the three classes, which are composed by 36% and 37% of particles, respectively, showed asymmetric conformations (referred to herein as conformations 1 and 2, respectively). In the middle region of each asymmetric dimer, two insulins bind at the adjacent site-1 and site-2', using the same interfaces as for the site-1 and site-2 insulin binding in the **T**-shaped IR-WT dimer (*Figure 4C*). Structural comparison revealed that the major structural differences between these two distinct asymmetric conformations arise from the different locations of insulin bound L1 domain in the middle part of the asymmetric dimer (*Figure 4D*). In the conformation 1, the L1 domain is located in a relatively lower position, close to the FnIII-2 domain (*Figure 4A, B and D*). In the conformation 2, the L1 domain rotates upward for approximately 30 degrees using the linker between the CR and L2 domains as a pivot point, resulting in a position close to the top part of the FnIII-1 domain (*Figure 4A, B and D*). In the meantime, the site-1 bound insulin also slides from the lower side of the site-2 insulin to the upper side. Additionally, in conformation 1 no contact is detected between the two insulins, whereas in conformation 2, the site-2 bound insulin weakly interacts with the site-1 bound insulin at the hybrid site (*Figure 4C*). Consistently, the insulin-bound L1 domain was better resolved in the cryo-EM map of conformation 2 than that of conformation 1.

We next compare the structure of asymmetric IR-3CS with that of asymmetric IRs determined at unsaturated insulin concentrations (*Li et al., 2022*; *Nielsen et al., 2022*; *Xiong et al., 2022*; *Figure 4—figure supplement 2*). All structures of asymmetric IR bound with unsaturated insulin exhibit similar structural features, that is, in the middle part of the asymmetric complex, one insulin bound at site-1 also makes weak contact with site-2' from the adjacent protomer or vice versa. However, in the asymmetric structure of IR-3CS with saturated insulin, two insulin molecules are bound at the hybrid site in the middle. To accommodate the binding of two insulins in the hybrid site, the L1/αCT together with bound site-1 insulin in the asymmetric IR-3CS moves outward as compared to the asymmetric IR-WT bound with unsaturated insulin. It is important to note that the outward movement of the L1 domain required for the binding of the second insulin to the hybrid site is made possible due to the physical decoupling of the αCTs by the 3CS mutation. This explains why the asymmetric conformation with four insulins bound was only observed in the cryo-EM dataset of IR-3CS/insulin.

The refinement of the third class from IR-3CS/insulin dataset, which comprises of 27% of the total good particles, resulted in a 3D reconstruction in a symmetric conformation at overall 3.7 Å resolution (*Figure 4A and B*). Four insulins are bound at sites-1 and –2 in the **T**-shaped dimer in a symmetric manner. A structural comparison of the **T**-shaped IR-3CS/insulin complex with the **T**-shaped IR-WT/

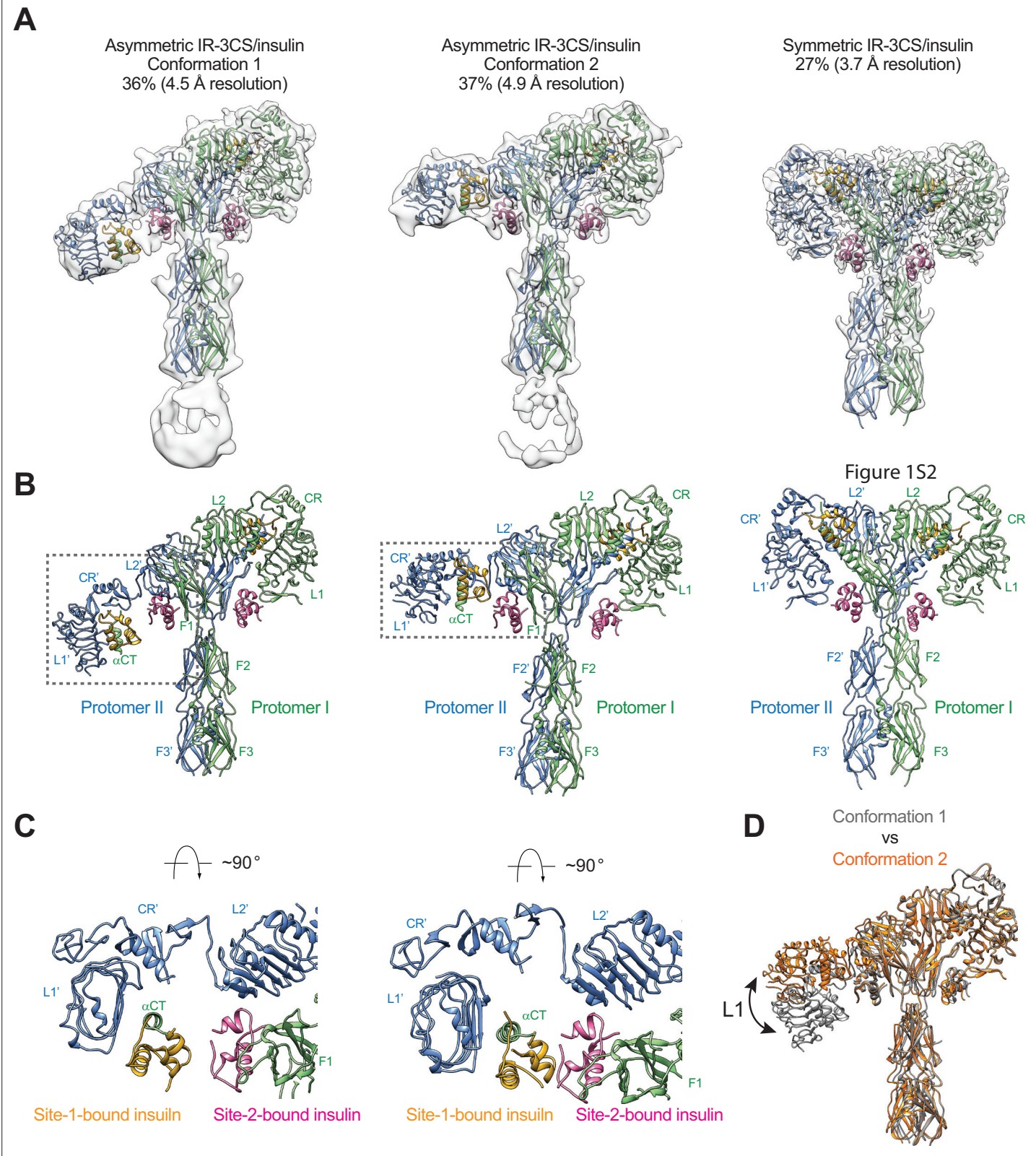

**Figure 4.** Structures of IR-3CS with insulin bound. (**A**) 3D reconstructions of the 2:4 IR-3CS/insulin complexes in both asymmetric and symmetric conformations, and the corresponding ribbon representation. The asymmetric conformations 1 and 2 were fitted into cryo-EM at 4.5 Å and 4.9 Å resolution, respectively. The symmetric conformation was fitted into cryo-EM at 3.7 Å resolution. The asymmetric conformations 1 and 2 were reconstructed from 36% and 37% of the well-defined particles, respectively. The symmetric conformation was reconstructed from 27% of the well-

*Figure 4 continued on next page*

*Figure 4 continued*

defined particles. (**B**) Ribbon representations of the asymmetric and symmetric IR-3CS/insulin complexes. (**C**) Top view of the asymmetric IR-3CS/insulin complexes. The location of this interaction in the asymmetric dimer is indicated by gray boxes in B. (**D**) Superposition between the asymmetric conformations 1 (gray) and 2 (orange), revealing conformational change of the L1/CR domains (indicated by the arrow).

The online version of this article includes the following figure supplement(s) for figure 4:

**Figure supplement 1.** Cryo-EM analysis of the IR-3CS/insulin complex.

**Figure supplement 2.** Structural comparison between asymmetric IR-3CS and asymmetric IR-WT with subsaturated insulin bound.

insulin complex revealed no significant structural differences (*Uchikawa et al., 2019*). However, whereas all the fully liganded IR-WT form the **T**-shaped symmetric conformation at saturated insulin concentrations, only a small portion of the fully liganded IR-3CS form the **T**-shaped dimer. These results indicate that the disulfide-linked αCT motifs play a critical role in promoting the **T**-shaped symmetric IR dimer.

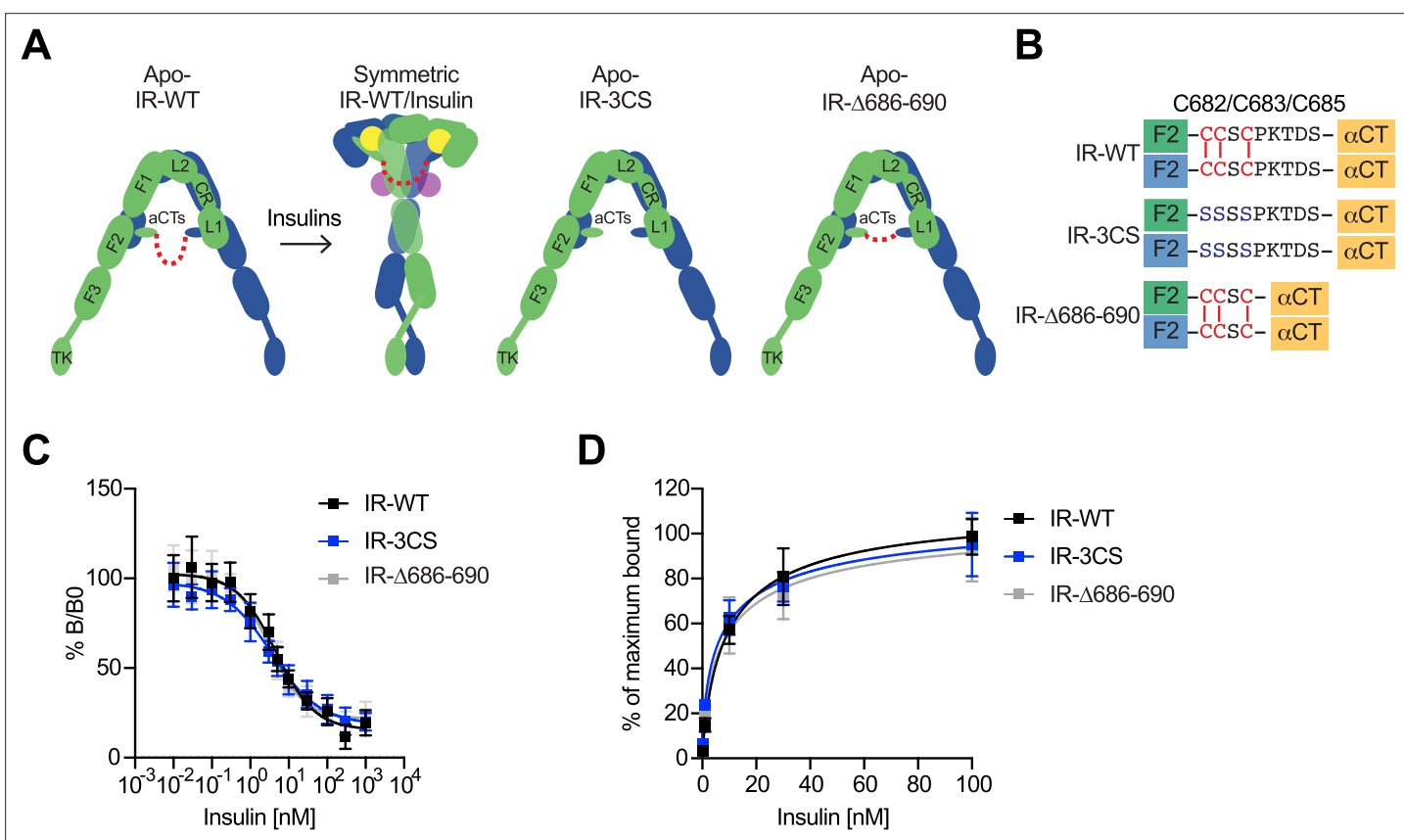

**Figure 5.** The function of disulfide-linked αCTs in the binding of insulin to IR. (**A**) Cartoon representations of the apo-IR, IR/insulin complex showing the IR dimer bound with four insulins (left), apo-IR-3CS (middle) and apo-IRΔ686–690 (right). The two IR protomers are colored in green and blue; the insulins at site-1 and site-2 are colored in yellow and purple, respectively. The linker between two αCTs is shown as red dotted line. (**B**) Sequences of the linker between two αCTs in the IR-WT, IR-3CS, and IR-Δ686–690 showing disulfide bonds in red. (**C**) Insulin competition-binding assay for full-length IR-WT, IR-3CS, and IR-Δ686–690. Mean ± SD. IR-WT, n=15 (IC$_{50}$=4.260 ± 0.9109 nM, Mean ± SD); IR-3CS, n=12 (IC$_{50}$=3.306 ± 0.4619 nM); IR-Δ686–690, n=12 (IC$_{50}$=3.231 ± 0.06734 nM). Source data are presented in *Figure 5—source data 1*. (**D**) Binding of insulin labeled with Alexa Fluor 488 to purified IR-WT, IR-3CS, and IR-Δ686–690 in the indicated conditions. Mean ± SD. n=24. Source data are presented in *Figure 5—source data 1*.

The online version of this article includes the following source data for figure 5:

**Source data 1.** Source data for *Figure 5C and D*.

## The disulfide-linked αCTs are required for optimal IR signaling and trafficking

To examine the importance of the disulfide-linked αCTs in the IR activation, we first tested the insulin binding to human IR-3CS (C682S/C683S/C685S) as well as IR-Δ686–690 (deletion of Pro689, Lys690, Thr691, Asp692, and Ser693), a mutant designed to have decreased flexibility of αCT motifs (*Figure 5A and B*). Unlike IGF1R, which shows increased IGF1 binding by physically decoupling the two αCTs (*Li et al., 2019*), alteration of the flexibility in the IR did not change the insulin binding properties (*Figure 5C and D*; *Figure 5—source data 1*). This result is consistent with the previous results using full-length IR with four glycine insertion in αCT (*Weis et al., 2018*).

We next analyzed insulin-dependent IR autophosphorylation (pY1150/1151 IR) and activating phosphorylation of protein kinase B (pAKT) and pERK in 293 FT cells expressing IR-WT or mutants at multiple time points (*Figure 6A and B*; *Figure 6—source data 1 and 2*). Insulin did not increase the levels of pY1150/1151 IR in cells expressing IR-Δ686–690, a mutant having much less flexible αCTs. Consistently, the levels of activating pAKT and pERK in cells expressing IR-Δ686–690 were only slightly increased upon insulin stimulation, suggesting that certain flexibility of disulfide-linked αCTs is required for robust IR activation. In contrast to IR-Δ686–690, the IR-3CS has a more flexible αCT, as its two αCTs are not physically linked by disulfide bonds. Although the pY1150/1151 level of IR-3CS was significantly increased upon insulin stimulation, the levels of pAKT and pERK were largely reduced in cells expressing IR-3CS. We speculate that the symmetric arrangement between the two membrane-proximal domains might be required for the robust downstream IR signaling. Therefore, the failure of IR-3CS to activate IR signaling may be due to the fact that only a small portion of IR-3CS could adopt a symmetric **T**-shaped conformation in response to insulin binding. These results further suggest that disulfide linkage between two αCTs is essential for the formation of the **T**-shaped IR dimer and normal IR signaling.

Insulin promotes clathrin-mediated IR endocytosis, which controls the intensity and duration of insulin signaling (*Goh and Sorkin, 2013*; *Najjar and Perdomo, 2019*; *Backer et al., 1990*; *Gustafson et al., 1995*). We and others have previously shown that mitotic regulators and SHP2 (SH2-containing protein tyrosine phosphatase 2)-MAPK (mitogen-activated protein kinase) pathway promote insulin-induced IR endocytosis (*Choi et al., 2019*; *Choi et al., 2016*; *Tang et al., 2020*). To study the contributions of the disulfide-linked αCTs to IR endocytosis, we generated HeLa cells stably expressing IR-GFP-WT, –3CS, or -Δ686–690, and examined the subcellular localization of these IR-GFP proteins (*Figure 6C and D*; *Figure 6—source data 2*). Without insulin stimulation, both IR-3CS and IR-Δ686–690 are localized to the plasma membrane, similar to IR-WT, suggesting that the introduced mutations do not induce protein mis-localization or misfolding. As expected, IR-WT underwent internalization upon insulin stimulation. Consistent with a role of IR kinase activation in IR endocytosis, IR-Δ686–690 did not undergo endocytosis upon insulin stimulation. Importantly, IR-3CS mutant was less efficiently internalized after insulin stimulation, potentially by defective activation of the MAPK pathway (*Choi et al., 2019*). These results suggest that enhancing structural flexibility of αCTs attenuates the formation of a stable **T**-shaped IR dimer, thus preventing normal IR signaling and trafficking.

## Discussion

The disulfide-linked αCTs is a unique structural motif that appears exclusively in IR family receptors. Our structural and functional results demonstrate that, although IR and IGF1R are closely related receptors, the disulfide-linked αCTs play distinct roles in their receptor activation.

### The functional role of disulfide-linked αCTs in the activation of IGF1R

In the structure of IGF1R-WT/IGF1, only one L1 domain is bound with IGF1, while the unliganded L1 plays a role in bridging the two membrane-proximal domains (*Figure 7A*; *Li et al., 2019*). Our new cryo-EM analysis shows that the IGF1R mutant that has more flexible disulfide-linked αCTs (IGF1R-P673G4) exhibits reduced negative cooperativity, which confirms our previous results that the disulfide-linked αCTs are the origin of the negative cooperativity in the binding of IGF1 to IGF1R. Furthermore, the binding of the second IGF1 to IGF1R-P673G4 delocalizes the L1 domain from the middle of the two membrane-proximal domains, and this IGF1-bound L1 undergoes free swing motion from lower to higher region of the receptor dimer using the linker between CR and L2 domains as the hinge

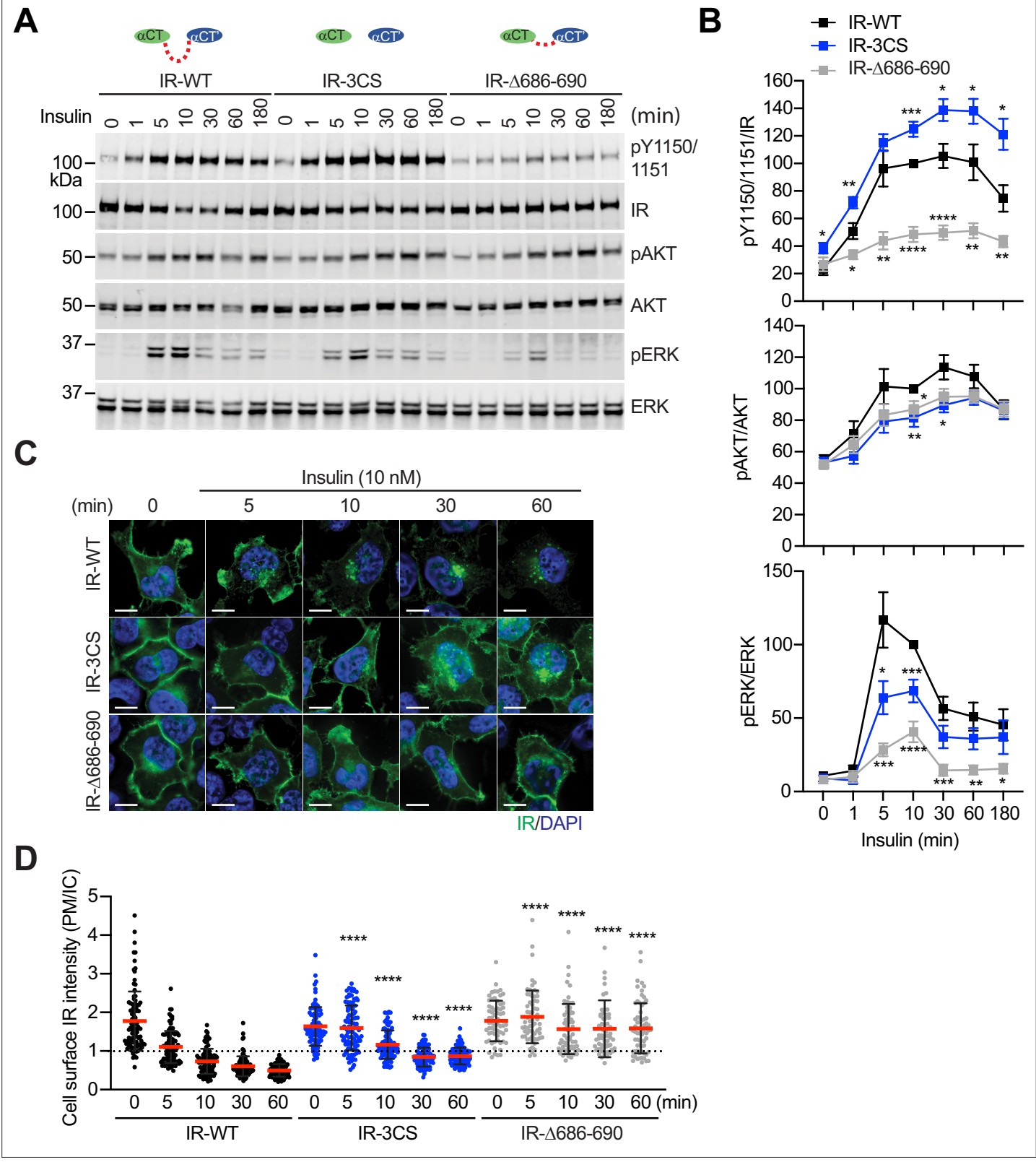

**Figure 6.** Functional importance of disulfide-linked αCTs on the IR activation. (**A**) Insulin-induced IR autophosphorylation, pAKT, and pERK levels in 293 FT cells expressing IR wild-type (WT), IR-3CS, and IR-Δ686–690. Cells were treated with 10 nM insulin for the indicated times. Cell lysates were blotted with the indicated antibodies. Source data are presented in *Figure 6—source data 1*. (**B**) Quantification of the western blot data shown in **A** (Mean ± SEM, pY IR/IR, WT, n=7; 3CS, n=13; Δ686–690, n=8; pAKT/AKT, WT, n=5; 3CS, n=7; Δ686–690, n=6; pERK/ERK, WT, n=7; 3CS, n=12; Δ686–

*Figure 6 continued on next page*

*Figure 6 continued*

690, n=9). Significance calculated using two-tailed student t-test; *p<0.05; **p<0.01; ***p<0.001; ****p<0.0001. Source data are presented in *Figure 6—source data 2*. (**C**) HeLa cells expressing IR-GFP WT, IR-GFP 3CS, or IR-GFP Δ686–690 were starved, treated with 10 nM insulin for indicated times, and stained with anti-GFP (IGF1R, green) and DAPI (blue). Scale bar, 10 μm. (**D**) Quantification of the ratios of plasma membrane (PM) and intracellular (IC) IGF1R-GFP signals of cells in **C** (WT0, n=97; WT5, n=102; WT10, n=98; WT30, n=88; WT60, n=103; 3CS0, n=88; 3CS5, n=98; 3CS10, n=92; 3CS30, n=93; 3CS60, n=94; Δ686-690-0, n=65; Δ686-690-5, n=63; Δ686-690-10, n=66; Δ686-690-30, n=64; Δ686-690-60, n=69). Mean ± SD; two-tailed student t-test; ****p<0.0001. Source data are presented in *Figure 6—source data 2*.

The online version of this article includes the following source data and figure supplement(s) for figure 6:

**Source data 1.** Source data for *Figure 6A*.

**Source data 2.** Source data for *Figure 6B and D*.

**Figure supplement 1.** The deletion of β-hairpin motif of L1 domain does not affect the IR activation.

**Figure supplement 1—source data 1.** Source data for *Figure 6—figure supplement 1B*.

**Figure supplement 1—source data 2.** Source data for *Figure 6—figure supplement 1C*.

(*Figure 7B*). As a result, the density for one of the two L1 domains was poorly resolved in most classes in the dataset of IGF1R-P673G4/IGF1. In the small population of particles, however, the L1 domain together with bound IGF1 moves to the top part of the receptor dimer, leading to a stable **T**-shaped symmetric conformation of the IGF1R with two IGF1s bound (*Figure 7B*). We noticed that the distance between the two membrane-proximal regions differs among apo-IGF1R, asymmetric IGF1R-WT/IGF1 and symmetric IGF1R-4G/IGF1 (65 Å, 39 Å and 55 Å, respectively) (*Figure 2C*). Notably, the distance between the two membrane-proximal domains of IGF1R-WT/IGF1 that is mediated by the unliganded

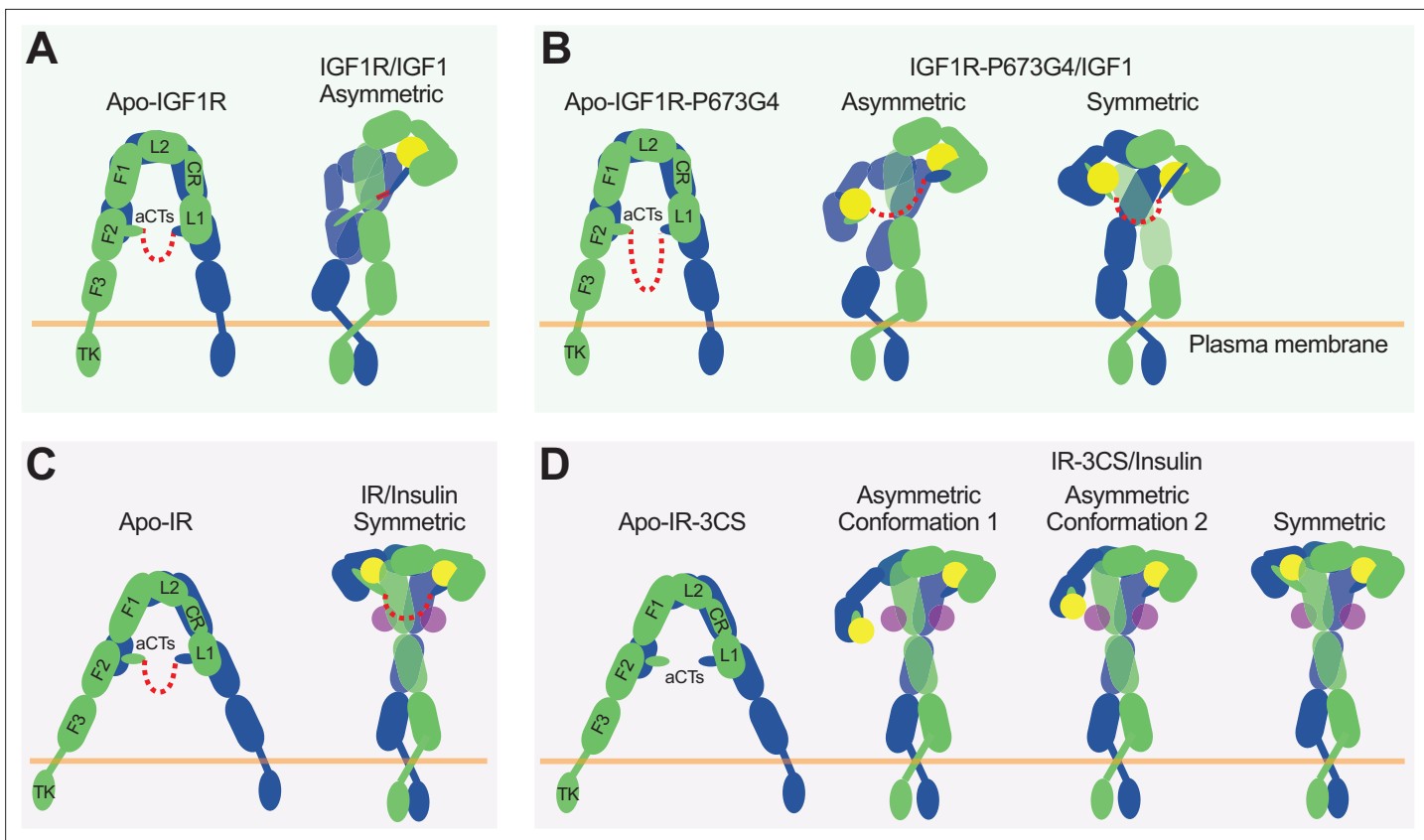

**Figure 7.** The activation mechanism of the IR and IGF1R by disulfide-linked αCTs. (**A**) IGF1-induced IGF1R-WT activation at saturated IGF1 concentrations. The two IGF1R protomers are colored in blue and green, respectively; the IGF1 is colored in yellow. (**B**) IGF1-induced IGF1R-P673G4 activation at saturated IGF1 concentrations. (**C**) Insulin-induced IR-WT activation at saturated insulin concentrations. The two IR protomers are colored in blue and green, respectively; insulins at site-1 and site-2 are colored in yellow and purple, respectively. (**D**) Insulin-induced IR-3CS activation at saturated insulin concentrations.

L1 domain is considerably shorter than that of IGF1R-P673G4/IGF1, which may favor efficient auto-phosphorylation of the intracellular kinase domains. Such structural differences explain partly why the IGF1R-WT exhibits higher kinase activity than IGF1R-P673G4 in the cell-based experiments.

Previous alanine scanning mutagenesis on IGF1 demonstrate that IGF1 mutations, including E9A, D12A, F16A, D53A, L54A, and E58A, markedly reduced IGF1R binding affinity (*Gauguin et al., 2008*). With the exception of IGF1 E9 that contacts site-1b of IGF1R (*Li et al., 2019*), none of IGF1 D12, F16, D53, L54, and E58 are involved in the binding to site-1, (*Alvino et al., 2011*) suggesting that IGF1 may have an additional binding site on IGF1R: It is worth to note that the IGF1 D12, F16, D53, L54, and E58 are equivalent residues to those of insulin that are important for the site-2 binding (*Uchikawa et al., 2019*). Despite saturating IGF1 levels, however, our previous and current structural studies did not reveal the putative site-2 of IGF1R (*Li et al., 2019*). We speculate that IGF1 may bind to the site-2 of IGF1R transiently during IGF1-induced IGF1R activation.

Altogether, our structural and functional studies demonstrate that disulfide-linked αCTs confer the negative cooperativity to the ligand binding in IGF1R. Ensured by the negative cooperativity, only one IGF1 is able to bind to one of the two site-1s in IGF1R, leading to a Γ-shaped asymmetric dimer which is the optimal conformation for IGF1R activation (*Figure 7A*). Therefore, our work illustrates the functional significance of negative cooperativity in IGF1-induced IGF1R activation.

## The functional role of αCT in the activation of IR

It has been shown that the binding of four insulins to two distinct sites of IR facilitates the full receptor activation (*Figure 7C*; *Li et al., 2022*). Previous structural analyses further indicate that IR predominantly forms asymmetric conformations when only one or two insulins are bound at IR (*Li et al., 2022*; *Weis et al., 2018*; *Nielsen et al., 2022*). In the two insulins bound asymmetric state, one insulin is bound at a hybrid site, simultaneously contacting site-1 and –2' from two adjacent protomers, and the disulfide-linked αCTs adopt a stretched conformation, due to the long distance between the two associated L1 domains (*Li et al., 2022*). The binding of another insulin to the hybrid site would require further outward movement of the L1 domain located in the hybrid site, leading to even more straightening of the disulfide-linked αCTs, which is likely to be energetically unfavorable. Nonetheless, it is reasonable to speculate that the more straightened disulfide-linked αCTs upon the binding of another insulin at the hybrid site would generate large contraction forces, thus promoting the conformational rearrangement of IR from asymmetric to symmetric. This explains why all the IR-WT particles would form a **T**-shaped symmetric conformation in response to saturated insulin concentrations (*Uchikawa et al., 2019*).

Removal of the disulfide bonds would physically decouple the two αCTs, allowing the binding of two insulins at the hybrid site without causing any stretching of the αCTs (*Figure 7D*). Indeed, in two of the three major classes from the dataset of IR-3CS/insulin, two insulins are observed at the hybrid site that bind at sites-1 and –2', respectively. Furthermore, the physical decoupling of the αCTs would eliminate its ability to trigger the conformational change of IR. This explains why in the IR-3CS/insulin complex, most particles adopt asymmetric conformations even at saturated insulin concentrations, which in turn further supports our hypothesis that both disulfide-linked αCTs and insulin binding to two distinct binding sites are critical for efficient formation of the **T**-shaped, active dimer of IR for the most optimal activation.

The membrane proximal regions of two protomers in the symmetric and asymmetric IR dimers are brough together which can undergo trans-autophosphorylation. Strikingly, our cellular functional assays demonstrate that the IR-3CS shows higher levels of autophosphorylation, but lower downstream signaling activity and defective endocytosis. We found that, although the membrane-proximal domain distances between asymmetric IR-3CS/insulin and symmetric IR-WT/insulin are similar, the relative position and orientation of the membrane-proximal domains between asymmetric and symmetric complexes differ significantly. Given the fact that the FnIII-3 domain is connected to the transmembrane domain by a short linker containing four amino acid residues, it is reasonable to speculate that extracellular, transmembrane, and intracellular domains are allosterically interconnected. Thus, the asymmetric arrangement of the membrane-proximal domains in the asymmetric IR-3CS/insulin complex may cause differential dimeric assembly of both transmembrane and intracellular domains compared to the symmetric IR/insulin complex, which might affect the interaction between the IR intracellular domains and downstream signaling regulators, leading to impaired downstream

signaling. Similar hypothesis has been proposed in the activation of EGFR (*Huang et al., 2021*). In addition, the endocytosis defect of IR-3CS may cause defective downstream signaling. Future studies are required to determine the structure of transmembrane and intracellular domains in the context of the entire full-length IR/insulin complex.

### The disulfide-linked αCTs play distinct roles in IR and IGF1R activation

The enhanced flexibility of the αCTs in IGF1R resulted in an almost twofold increase in the binding of IGF1 to the IGF1R (*Li et al., 2019*). The binding of the second IGF1 to IGF1R disrupts the stable Γ-shaped dimer, leading to suboptimal IGF1R activation. In contrast, the enhanced flexibility of the αCTs in IR did not alter the characteristics of insulin binding but greatly diminished the ability of IR in forming a **T**-shaped IR dimer as well as potently activating the downstream signaling. These data suggest that the αCTs regulate the ligand binding and receptor activation in distinct ways between IR and IGF1R.

The remaining question is that how these two related receptors are activated in such distinct ways, that is IGF1R activated by a single IGF1 adopts the Γ-shaped asymmetric conformation meanwhile IR activated by multiple insulin molecules forms the **T**-shaped symmetric conformation. The IR and IGF1R share a high degree of structural similarity, however, sequences in their extracellular domains are less conserved (*Figure 1—figure supplement 1*), and the characteristics of ligand binding are significantly different between two receptors. For example, a β-hairpin motif containing residues 163–174 of the L1 domain of IGF1R stabilizes the one IGF1 bound asymmetric active dimer by contacting the FnIII-2 domain of the same protomer (*Figure 1C*; *Li et al., 2019*). The deletion of the entire L1 β-hairpin (Δ163–174) significantly reduced the IGF1-dependent activation of IGF1R (*Li et al., 2019*). The key residues for the intra L1-FnIII-2 interaction, such as K164 and Y173, are not conserved in the IR (*Figure 6—figure supplement 1A*). Indeed, deletion of the L1 β-hairpin of IR (Δ170–181, corresponding to IGF1R residence 163–174) did not affect the insulin-dependent IR activation (*Figure 6—figure supplement 1B and C*; *Figure 6—figure supplement 1—source data 1 and 2*). This suggests that the αCT and other extracellular regions of the receptors contribute to the distinct active conformations between IR and IGF1R.

Previous kinetics assays for negative cooperativity demonstrate that the significant difference between IR and IGF1R for ligand binding lies in the fact that IR's negative cooperativity diminishes progressively when insulin concentrations exceed 100 nM, whereas the IGF1R maintains its maximal negative cooperativity at saturated levels of IGF1 (*Kiselyov et al., 2009*; *De Meyts, 1994*). This is consistent with our structural analysis that the IR predominantly adopts a symmetrical conformation with multiple insulins bound, while the IGF1R adopts an asymmetric conformation with a single IGF1 bound, even at saturated IGF1 concentrations. The circulating levels of insulin are rapidly oscillating depending on food intake, which mainly mediate metabolic activity. In contrast, the IGF1 levels remain relatively stable during the daytime and control long-term actions such as cell growth and differentiation, even though they share common signaling pathways (*Siddle, 2012*; *Dupont and LeRoith, 2001*; *Kim and Accili, 2002*). Given the nature of these differences between insulin and IGF1, the distinct roles of disulfide-linked αCTs in the receptor activation could contribute to biological outcomes of the IR family signaling. Future studies are required to determine whether these structural differences modulate the association with downstream effectors and adaptors, contributing to further signaling pathways.

## Materials and methods

All reagents generated in this study are available with a completed Materials Transfer Agreement.

### Cell lines

#### FreeStyle™ 293-F

FreeStyle™ 293 F cells were obtained from Invitrogen (R79007). FreeStyle™ 293 F cells were cultured in FreeStyle 293 Expression Medium. FreeStyle™ 293 F cells were maintained in orbital shaker in 37 °C incubator with a humidified atmosphere of 5% $CO_2$.

## HEK293S GnTI-

HEK293S GnTI- cells were obtained from ATCC (CRL-3022). The cells were cultured in FreeStyle 293 Expression Medium and maintained in orbital shaker in 37 °C incubator with a humidified atmosphere of 5% $CO_2$.

## 293FT

293 FT cells were obtained from Invitrogen (R70007). 293 FT cells were cultured in high-glucose Dulbecco's modified Eagle's medium (DMEM) supplemented with 10% (v/v) fetal bovine serum (FBS), 2 mM L-glutamine, and 1% penicillin/streptomycin. 293 FT cells were maintained in monolayer culture at 37 °C and 5% $CO_2$ incubator.

## HeLa Tet-on

HeLa Tet-on cells were obtained from Takara Bio. HeLa Tet-on cells were cultured in high-glucose DMEM supplemented with 10% (v/v) FBS, 2 mM L-glutamine, and 1% penicillin/streptomycin. HeLa Tet-on cells were maintained in monolayer culture at 37 °C and 5% $CO_2$ incubator.

## Sf9 cells

*Spdoptera frugiperda* (Sf9) cells were cultured in SF900 II SFM (Gibco) at 27 °C with orbital shaking at 120 rpm.

## Cell line validation

An aliquot of each cell line was passaged for only 3–4 weeks, after which a fresh batch of cells was thawed and propagated. No mycoplasma contaminations were detected.

## Protein expressing and purification

Mature human IGF1 gene (residues 49–118) was subcloned into modified pET-28a vector encoding a His6-tag and SUMO-tag at N-terminus. The protein was expressed as inclusion bodies in *E. coli* strain BL21 (DE3). The bacteria pellet was suspended in buffer containing 50 mM Tris-HCl pH 8.0, 2 mM EDTA. After the bacteria was disrupted, the inclusion body was washed and dissolved in buffer containing 8 M urea, 30 mM Tris-HCl pH 8.0, 1 mM EDTA and 5 mM dithiothreitol (DTT). After centrifuge at 20,000 g for 30 min, the solubilized inclusion body was refolded by dialysis method against buffer 500 mM arginine pH 8.0, 0.6 mM oxidized glutathione overnight at room temperature. The human IGF1 was then purified by Ni column and eluted with ULP1 enzyme cleavage. The protein was further purified with superdex 75 increase 10/300 GL size-exclusion column (Cytiva) in the buffer 20 mM Hepes-Na pH 7.4, 150 mM NaCl.

The amino acid numbering for IGF1R and IR starts after the signal peptide. Protein expression and purification were performed following previous protocols (*Li et al., 2022*). Briefly, for the expression of MmIGF1R-P673G4, the coding sequence of full-length mouse IGF1R (NM_010513.2) with C-terminal tail truncation (residues 1–1262), kinase dead mutation (D1107N), endocytosis defective mutation (Y951A) and four glycine insertion between P673 and K674 was subcloned into pEZT-BM vector. For the expression of MmIR-3CS, the coding sequence of full-length mouse IR (NM_010568.3) with kinase dead mutation (D1122N), endocytosis defective mutation (Y962F) and cysteine mutation (C684S, C685S, C687S) was subcloned into pEZT-BM vector. Both MmIGF1R-P673G4 and MmIR-3CS were purified with Tsi3 tag based on the high affinity binding of Tse3 and Tsi3 protein. HRV-3C protease cleavage sequence was introduced between the receptor gene and Tsi3 gene. Both proteins were expressed with Bac-to-Bac system. Briefly, the fusion constructs were transformed to DH10Bac strain to produce bacmids. Bacmids were transfected into sf9 insect cells to produce baculovirus. Then baculovirus at 1:10 (v:v) ratio were used to infect mammalian cells to express protein. MmIGF1R-P673G4 (note that we used the amino acid numbering of human IGF1R and IR to describe our structural and functional analysis) was expressed in HEK293S GnTI- cells (ATCC, CRL-3022), while MmIR-3CS was expressed in FreeStyle293-F cells (Invitrogen, R79007). A total of 10 mM sodium butyrate was added to the medium to boost expression. MmIGF1R-P673G4 was expressed at 30 °C while MmIR-3CS was expressed at 37 °C. The cells were grown for 48 hr before harvesting.

All purification step was carried out at 4 °C. The cells expressing MmIGF1R-P673G4 or MmIR-3CS were resuspended and lysed by cell disruptor in the buffer containing 40 mM Tris-HCl, pH 8.0, 400 mM

NaCl and protease inhibitor cocktail (Roche). The membrane fraction was pelleted by ultracentrifuge at 100,000 g for 1 hr. 1% (w/v) n-Dodecyl-β-D-Maltopyranoside (DDM) (Anatrace) was added with stirring overnight to extract the proteins from the membrane fraction. After ultracentrifuge for 1 hr at 100,000 g, the supernatant was supplemented with 1 mM $CaCl_2$ at final concentration and loaded to Tse3 conjugated Sepharose resin (Cytiva) by gravity flow. The resin was washed by 20 column volumes of buffer 50 mM Tris-HCl, pH 8.0, 400 mM NaCl, 1 mM $CaCl_2$, 5% glycerol, 0.08% DDM. MmIGF1R-P673G4 or MmIR-3CS was then eluted from the resin by incubating with HRV-3C protease for 3 hr. The protein without tag was further purified by size-exclusion column Superose 6 increase 10/300 GL (Cytiva) in the buffer 20 mM Hepes-Na pH 7.4, 150 mM NaCl, 0.03% DDM. The dimer fraction was pooled and incubated with human IGF1 at 1:1 (m:m) or human insulin (sigma) at 1:4 (m:m) for 1 hr at 4 °C. The protein was further concentrated to 4–6 mg/ml for cryo-EM analyses.

## EM data acquisition

EM data acquisition, image processing, and model building, and refinement were performed following previous protocols with some modifications (*Li et al., 2022*).

The cryo-EM grid was prepared by applying 3 μl of the protein samples (4–6 mg/ml) to glow-discharged Quantifoil R1.2/1.3 300-mesh gold holey carbon grids (Quantifoil, Micro Tools GmbH, Germany). Grids were blotted for 4.0 s under 100% humidity at 4 °C before being plunged into the liquid ethane using a Mark IV Vitrobot (FEI). Micrographs were acquired on a Titan Krios microscope (FEI) operated at 300 kV with a K3 direct electron detector (Gatan), using a slit width of 20 eV on a GIF-Quantum energy filter. SerialEM 3.8 was used for the data collection. A calibrated magnification of 60,241 was used for imaging of MmIGF1R-P673G4/IGF1 samples, yielding a pixel size of 0.83 Å on specimen. A calibrated magnification of 46,296 was used for imaging of MmIR-3CS/insulin samples, yielding a pixel size of 1.08 Å on specimen. The defocus range was set from 1.6 μm to 2.6 μm. Each micrograph was dose-fractionated to 30 frames with a total dose of about 60 e$^-$/Å$^2$.

## Image processing

The cryo-EM refinement statists for 2 datasets are summarized in *Table 1*. 3,986 movie frames of IGF1R-P673G4/IGF1 micrographs were motion-corrected and binned two-fold, resulting in a pixel size of 0.83 Å, and dose-weighted using MotionCor2. The CTF parameters were estimated using Gctf. RELION3 was used for the following processing. Particles were first roughly picked by using the Laplacian-of-Gaussian blob method, and then subjected to 2D classification. Class averages representing projections of the IGF1R-P673G4/IGF1 in different orientations were used as templates for reference-based particle picking. Extracted particles were binned three times and subjected to 2D classification. Particles from the classes with fine structural feature were selected for 3D classification using an initial model generated from a subset of the particles in RELION. Particles from one of the resulting 3D classes showing good secondary structural features were selected and re-extracted into the original pixel size of 1.08 Å. Subsequently, we performed finer 3D classification by using local search in combination with small angular sampling, resulting one class showing C2 symmetric conformation. All the other classes were all resolved as asymmetric conformation, with one of the two L1 domains completely flexible. The cryo-EM map for the symmetric class after 3D refinement with C2 symmetry imposed was resolved at 4 Å resolution.

7,633 movie frames of IR-3CS/insulin micrographs were motion-corrected and binned two-fold, resulting in a pixel size of 1.08 Å, and dose-weighted using MotionCor2. CTF correction was performed using Gctf. Particles were first roughly picked by using the Laplacian-of-Gaussian blob method, and then subjected to 2D classification. Class averages representing projections of the IR-3CS/insulin complex in different orientations were used as templates for reference-based particle picking. A total of 3,283,617 particles were picked. Particles were extracted and binned by three times (leading to 3.24 Å/pixel) and subjected to another round of 2D classification. Particles in good 2D classes were chosen (3,128,358 in total) for 3D classification using an initial model generated from a subset of the particles in RELION3. After initial 3D classification IR-3CS/insulin particles set, three major classes were identified showing good secondary structural features (asymmetric and symmetric, respectively). Particles for two different asymmetric conformations or one symmetric conformation were selected separately and re-extracted into the original pixel size of 1.08 Å. The final reconstructions of two

**Table 1.** Cryo-EM data collection and refinement statistics.

| | IGF1R-P673G4/IGF1 Symmetric | IR-3CS/insulin Symmetric | IR-3CS/insulin Asymmetric conformation 1 | IR-3CS/insulin Asymmetric conformation 2 |
|---|---|---|---|---|
| **Data collection and processing** | | | | |
| Magnification | 60,241 | 46,296 | 46,296 | 46,296 |
| Voltage (kV) | 300 | 300 | 300 | 300 |
| Electron exposure ($e^-/Å^2$) | 60 | 60 | 60 | 60 |
| Defocus range (μm) | 1.6–2.6 | 1.6–2.6 | 1.6–2.6 | 1.6–2.6 |
| Pixel size (Å) | 0.83 | 1.08 | 1.08 | 1.08 |
| Symmetry imposed | C2 | C2 | C1 | C1 |
| Map resolution (Å) | 4.0 | 4.5 | 4.9 | 3.7 |
| FSC threshold | 0.143 | 0.143 | 0.143 | 0.143 |
| | | | | |
| **Refinement** | | | | |
| Initial model used (PDB code) | 6PYH | 6PXV | 6PXV | 6PXV |
| Model composition | | | | |
| Nonhydrogen atoms | 13,596 | 14,640 | 14,108 | 14,100 |
| Protein residues | 1,702 | 1,815 | 1,748 | 1,747 |
| Ligands | 0 | 0 | 0 | 0 |
| R.m.s. deviations | | | | |
| Bond lengths (Å) | 0.005 | 0.004 | 0.004 | 0.004 |
| Bond angles (°) | 1.091 | 0.987 | 0.978 | 0.957 |
| | | | | |
| **Validation** | | | | |
| MolProbity score | 2.49 | 2.01 | 2.14 | 2.37 |
| Clashscore | 25.81 | 12.03 | 15.55 | 26.58 |
| Poor rotamers (%) | 0.74 | 0.18 | 0.06 | 0.06 |
| Ramachandran plot | | | | |
| Favored (%) | 88.48 | 93.61 | 93.17 | 92.72 |
| Allowed (%) | 11.28 | 6.27 | 6.65 | 7.16 |
| Disallowed (%) | 0.24 | 0.11 | 0.18 | 0.12 |

asymmetric-complexes and symmetric-complex were resolved at 4.5 Å, 4.9 Å, and 3.7 Å resolution, respectively.

## Model building and refinement

Model buildings of IGF1R-P673G4/IGF1 and IR-3CS/insulin were initiated by rigid-body docking of individual domains from the cryo-EM/crystal structures of L1, CR, L2, and FnIII1-3 domains of IGF1R and IR, IGF1 and insulin. Manual building was carried out using the program Coot. The model was refined by using the real-space refinement module in the Phenix package (V1.17). Restraints on secondary structure, backbone Ramachandran angles, residue sidechain rotamers were used during

the refinement to improve the geometry of the model. MolProbity 4.5 as a part of the Phenix validation tools was used for model validation (*Table 1*). Figures were generated in Chimera 1.16.

## IR and IGF1R activation and signaling assay in cultured cells

The IR and IGF1R signaling assay were performed as described earlier with some modifications (*Uchikawa et al., 2019*; *Li et al., 2022*; *Li et al., 2019*; *Park et al., 2022*). For the activation assay, the short isoform of human IR and human IGF1R in pCS2-MYC were used as described previously (*Uchikawa et al., 2019*; *Li et al., 2019*; *Choi et al., 2016*). Mutants of IR and IGF1R were generated by Q5 site-directed mutagenesis kit (New England BioLabs Inc). Plasmid transfections into 293 FT cells were performed with Lipofectamin 2000 (Invitrogen). One day later, the cells were serum starved for 14–16 hr and treated 10 nM insulin or 50 nM IGF1 for the indicated time points.

The cells were incubated with the lysis buffer B [50 mM Hepes pH 7.4, 150 mM NaCl, 10% Glycerol, 1% Triton X-100, 1 mM EDTA, 100 mM sodium fluoride, 2 mM sodium orthovanadate, 20 mM sodium pyrophosphate, 0.5 mM dithiothreitol (DTT), 2 mM phenylmethylsulfonyl fluoride (PMSF)] supplemented with cOmplete Protease Inhibitor Cocktail (Roche) and PhosSTOP (Sigma) on ice for 1 hr. After centrifuge at 20,817 g at 4 °C for 10 min, the concentrations of cell lysate were measured using Micro BCA Protein Assay Kit (Thermo Scientific). About 30–40 µg total proteins were analyzed by SDS-PAGE and quantitative Western blotting. The following antibodies were purchased from commercial sources: Anti-IR-pY1150/1151 (WB, 1:1000; 19H6; labeled as pY IGF1R or pY 1150/1151, Cat. #3024), anti-AKT (WB, 1:1000; 40D4, Cat. #2920), anti-pS473 AKT (WB, 1:1000; D9E, Cat. #4060), anti-ERK1/2 (WB, 1:1000; L34F12, Cat. #4696), anti-pERK1/2 (WB, 1:000; 197G2, Cat. #4377, Cell Signaling); anti-Myc (WB, 1:2000; 9E10, Cat. #11667149001, Roche; labeled as IR or IGF1R). Generation and validation of rabbit polyclonal antibodies against GFP was described previously (*Xia et al., 2004*; *Tang et al., 2001*). For quantitative western blots, anti-rabbit immunoglobulin G (IgG) (H+L) (Dylight 800 conjugates, Cat. #5151) and anti-mouse IgG (H+L) (Dylight 680 conjugates, Cat. #5470) (Cell Signaling) were used as secondary antibodies. The membranes were scanned with the Odyssey Infrared Imaging System (Li-COR, Lincoln, NE). Levels of phosphorylation were normalized to total protein and shown as intensities relative to that in cells treated with 10 nM insulin for 10 min or to that in cells treated with 50 nM IGF1 for 10 min.

## Immunofluorescence assay for IR and IGF1R trafficking

The IR and IGF1R trafficking assay were performed as described earlier with some modifications (*Choi et al., 2019*; *Choi et al., 2016*). For the generation of IR-GFP or IGF1R-GFP expressing HeLa Tet-on cell lines, cDNAs encoding human IR or IGF1R were cloned into the pBabe-puro vector. pBabe-puro-IR or pBabe-puro-IGF1R vectors were co-transfected with p-gag/pol and pCMV-VSV-G into 293 FT cells using Lipofectamin 2000 (Invitrogen). Viral supernatants were collected at 2 and 3 days after transfection, concentrated with Retro X concentrator (Clonetech), and then added to HeLa Tet-on with polybrene (4 µg/ml). Cells were treated with 2 µg/ml of puromycin at 3 days after infection and selected for 2 weeks. The cells were plated on coverslips, starved for 14 hr, and then treated with 10 nM insulin or 50 nM IGF1 for the indicated times. Cells were fixed with cold methanol for 10 min, washed with 0.1% Triton X-100 in PBS (0.1% PBST) and incubated with 3% bovine serum albumin (BSA) in 0.1% PBST for 1 hr. The cells were incubated with diluted anti-GFP antibodies (1:500) in 3% BSA in 0.1% PBST overnight at 4 °C. After wash, the cells were incubated with fluorescent secondary antibodies (1:200, Alexa fluor 488 goat anti-rabbit IgG (H+L) antibody, Cat. #A11008, Invitrogen) and mounted on microscope slides in ProLong Gold Antifade reagent with DAPI (Invitrogen). Images were acquired with a Leica thunder Imager (Leica Microsystems Inc). The cell edges were defined with Image J. The whole cell (WC) signal intensity and intracellular (IC) signal intensity were measured. The plasma membrane (PM) signal intensity was calculated by subtracting IC from WC. Identical exposure times and magnifications were used for all comparative analyses.

## In vitro insulin-binding assay

In vitro insulin-binding assay was conducted as previously described with slight modification (*Li et al., 2022*). To isolate IR-WT, −3CS, -Δ686–690 proteins, 239 FT cells were transfected with pCS2-IR-Myc-WT, −3CS, or -Δ686–690 using Lipofectamin 2000 (Invitrogen). Two days later, the cells were serum starved for 14 hr. The cells were lysed with lysis buffer B without DTT supplemented with cOmplete Protease

Inhibitor cocktail (Roche) and PhosSTOP (Sigma) on ice for 1 hr. After centrifugation at 20,817 g at 4 °C for 10 min, the concentrations of cell lysate were measured using Micro BCA Protein Assay Kit (Thermo Fisher Scientific). Cell lysates and anti-c-Myc magnetic beads (Cat. #88842, Thermo Fisher Scientific, 250 µg of beads per 3 mg of total cell lysates) were incubated at 4 °C for 2 hr. The beads were washed two times with the washing buffer B [50 mM Hepes pH 7.4, 400 mM NaCl, 0.05% NP-40] supplemented with cOmplete Protease Inhibitor cocktail (Roche) and PhosSTOP (Sigma). The beads were washed once with binding buffer [20 mM Hepes pH 7.4, 200 mM NaCl, 0.03% Dodecyl malto-side (DDM), and 0.003% cholesteryl hemisuccinate (CHS) (Anatrace)] supplemented with cOmplete Protease Inhibitor cocktail (Roche) and 100 nM BSA and resuspended in 25 µl of the binding buffer. Seven µl of IR-bound beads and the indicated amount of Alexa Fluor labeled insulin analogs were incubated on a rotator at 4 °C for 1 hr. For competition binding assay, 10 µl of IR-bound beads, 0.5 nM Alexa Fluor labeled human insulin (*Uchikawa et al., 2019*; *Li et al., 2022*), and the indicated amount of insulin were incubated on a rotator at 4 °C for 14 hr. The beads were washed once with the binding buffer. The bound proteins were eluted with 50 µl of binding buffer containing 2% SDS at 50 °C for 10 min. The samples were diluted with 150 µl of binding buffer. The fluorescence intensities were measured in a microplate reader (Cytation 5; Biotek). Non-specific binding was measured in samples of Alexa Fluor labeled insulin analogs with beads without IR and subtracted from the data.

## Statistical analysis

Prism 9 was used for the generation of graphs and for statistical analyses. Results are presented as mean ± s.d. or mean ± s.e.m. Two-tailed unpaired $t$ tests were used for pairwise significance analysis. Power analysis for sample sizes were not performed. Randomization and blinding methods were not used, and data were analyzed after the completion of all data collection in each experiment.

## Acknowledgements

Cryo-EM data were collected at the University of Texas Southwestern Medical Center (UTSW) Cryo-Electron Microscopy Facility, funded in part by the Cancer Prevention and Research Institute of Texas (CPRIT) Core Facility Support Award RP170644. We thank Dr. Stoddard for facility access, Dr. Julie Canman for helpful discussion, and Dr. Youshin Suh for sharing the plate reader. This work is supported in part by grants from the National Institutes Health (R35GM142937 to EC; R01GM136976 to X.-CB), Columbia Digestive and Liver Diseases Research Center Pilot grant (1P30DK132710 to EC), the Welch foundation (I-1944 to X-CB), CPRIT (RP160082 to X-CB), and the Alice Bohmfalk Charitable (to EC). X-CB is Virginia Murchison Linthicum Scholar in Medical Research at UTSW.

## Additional information

### Funding

| Funder | Grant reference number | Author |
| --- | --- | --- |
| National Institute of General Medical Sciences | R35GM142937 | Eunhee Choi |
| National Institute of General Medical Sciences | R01GM136976 | Xiao-chen Bai |
| National Institute of Diabetes and Digestive and Kidney Diseases | 1P30DK132710 | Eunhee Choi |
| Welch Foundation | I-1944 | Xiao-chen Bai |
| Cancer Prevention and Research Institute of Texas | RP160082 | Xiao-chen Bai |
| Alice Bohmfalk Charitable | | Eunhee Choi |
| Virginia Murchison Linthicum Scholar in Medical Research at UTSW | | Xiao-chen Bai |

| Funder | Grant reference number | Author |
|---|---|---|

The funders had no role in study design, data collection and interpretation, or the decision to submit the work for publication.

## Author contributions

Jie Li, Jiayi Wu, Catherine Hall, Data curation, Formal analysis, Writing – review and editing; Xiao-chen Bai, Eunhee Choi, Conceptualization, Data curation, Formal analysis, Supervision, Funding acquisition, Validation, Investigation, Visualization, Methodology, Writing - original draft, Project administration, Writing – review and editing

## Author ORCIDs

Jie Li http://orcid.org/0000-0002-1059-280X
Jiayi Wu http://orcid.org/0000-0002-9692-2864
Xiao-chen Bai http://orcid.org/0000-0002-4234-5686
Eunhee Choi http://orcid.org/0000-0003-3286-6477

## Decision letter and Author response

Decision letter https://doi.org/10.7554/eLife.81286.sa1
Author response https://doi.org/10.7554/eLife.81286.sa2

# Additional files

## Supplementary files

• Transparent reporting form

## Data availability

All cryo-EM maps and models reported in this work has been deposited into EMDB/PDB database under the entry ID: PDB-8EYR and EMD-28693 (IGF1R/IGF1, symmetric conformation), PDB-8EYX and EMD-28723 (IR-3CS/insulin, asymmetric conformation 1), PDB-8EYY and EMD-28724 (IR-3CS/insulin, asymmetric conformation 2), and PDB-8EZ0 and EMD-28725 (IR-3CS/insulin, symmetric conformation). Source data are provided with this paper.

The following datasets were generated:

| Author(s) | Year | Dataset title | Dataset URL | Database and Identifier |
|---|---|---|---|---|
| Li J, Wu J, Hall C, Bai XC, Choi E | 2022 | Cryo-EM structure of two IGF1 bound full-length mouse IGF1R mutant (four glycine residues inserted in the alpha-CT; IGF1R-P674G4): symmetric conformation | https://www.rcsb.org/structure/8EYR | RCSB Protein Data Bank, PDB-8EYR |
| Li J, Wu J, Hall C, Bai XC, Choi E | 2022 | Cryo-EM structure of two IGF1 bound full-length mouse IGF1R mutant (four glycine residues inserted in the alpha-CT; IGF1R-P674G4): symmetric conformation | https://www.ebi.ac.uk/emdb/EMD-28693 | Electron Microscopy Data Bank, EMD-28693 |
| Li J, Wu J, Hall C, Bai XC, Choi E | 2022 | Cryo-EM structure of 4 insulins bound full-length mouse IR mutant with physically decoupled alpha CTs (C684S/C685S/C687S; denoted as IR-3CS) Asymmetric conformation 1 | https://www.rcsb.org/structure/8EYX | RCSB Protein Data Bank, PDB-8EYX |

*Continued*

| Author(s) | Year | Dataset title | Dataset URL | Database and Identifier |
|---|---|---|---|---|
| Li J, Wu J, Hall C, Bai XC, Choi E | 2022 | Cryo-EM structure of 4 insulins bound full-length mouse IR mutant with physically decoupled alpha CTs (C684S/C685S/C687S; denoted as IR-3CS) Asymmetric conformation 1 | https://www.ebi.ac.uk/emdb/EMD-28723 | Electron Microscopy Data Bank, EMD-28723 |
| Li J, Wu J, Hall C, Bai XC, Choi E | 2022 | Cryo-EM structure of 4 insulins bound full-length mouse IR mutant with physically decoupled alpha CTs (C684S/C685S/C687S, denoted as IR-3CS) Asymmetric conformation 2 | https://www.rcsb.org/structure/8EYY | RCSB Protein Data Bank, PDB-8EYY |
| Li J, Wu J, Hall C, Bai XC, Choi E | 2022 | Cryo-EM structure of 4 insulins bound full-length mouse IR mutant with physically decoupled alpha CTs (C684S/C685S/C687S, denoted as IR-3CS) Asymmetric conformation 2 | https://www.ebi.ac.uk/emdb/EMD-28724 | Electron Microscopy Data Bank, EMD-28724 |
| Li J, Wu J, Hall C, Bai XC, Choi E | 2022 | Cryo-EM structure of 4 insulins bound full-length mouse IR mutant with physically decoupled alpha CTs (C684S/C685S/C687S; denoted as IR-3CS) Symmetric conformation | https://www.rcsb.org/structure/8EZ0 | RCSB Protein Data Bank, PDB-8EZ0 |
| Li J, Wu J, Hall C, Bai XC, Choi E | 2022 | Cryo-EM structure of 4 insulins bound full-length mouse IR mutant with physically decoupled alpha CTs (C684S/C685S/C687S; denoted as IR-3CS) Symmetric conformation | https://www.ebi.ac.uk/emdb/EMD-28725 | Electron Microscopy Data Bank, EMD-28725 |

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

# Appendix 1

## Appendix 1—key resources table

| Reagent type (species) or resource | Designation | Source or reference | Identifiers | Additional information |
|---|---|---|---|---|
| Strain, strain background (*Escherichia coli*) | One Shot Stbl3 Chemically Competent *E. coli* | Life Technologies | C7373-03 | |
| Strain, strain background (*Escherichia coli*) | BL21 (DE3) | Life Technologies | C600003 | |
| Strain, strain background (*Escherichia coli*) | MAX Efficiency DH10Bac Competent Cells | Thermo Fisher | 10361012 | |
| Cell line (*Homo-sapiens*) | 293 FT | Invitrogen | R70007 | |
| Cell line (*Homo-sapiens*) | FreeStyle 293 F | Invitrogen | R79007 | |
| Cell line (*Homo-sapiens*) | HEK293S GnTI- | ATCC | CRL-3022 | |
| Cell line (*Homo-sapiens*) | HeLa Tet-on | Takara Bio | 631183 | |
| Cell line (*Spdoptera frugiperda*) | Sf9 | ATCC | CRL-1711 | |
| Transfected construct (*Homo-sapiens*) | pET-28a-His6-SUMO-IGF1 | This paper | | Protein expressing and purification: Mature human IGF1 gene (residues 49–118) was subcloned into modified pET-28a vector encoding a His6-tag and SUMO-tag at N-terminus. |
| Transfected construct (*Mus musculus*) | pEZT-BM-mouse IGF1R-P673G4 | This paper | | Protein expressing and purification: NM_010513.2 with C-terminal tail truncation (residues 1–1262), D1107N, Y951A, and four glycine insertion between P673 and K674 |
| Transfected construct (*Mus musculus*) | pEZT-BM-mouse insulin receptor (IR)–3CS | This paper | | Protein expressing and purification: NM_010568.3 with D1122N, Y962F, C684S, C685S, and C687S |
| Transfected construct (*Homo-sapiens*) | pCS2-human IR WT-MYC | *Choi et al., 2016*, Cell | | |
| Transfected construct (*Homo-sapiens*) | pCS2-human IR-3CS-MYC | This paper | | Cysteine mutation (C682S, C683S, C685S) |
| Transfected construct (*Homo-sapiens*) | pCS2-human IR Δ686–690-MYC | This paper | | Deletion residues 686–690 |
| Transfected construct (*Homo-sapiens*) | pCS2-human IR Δ170–181-MYC | This paper | | Deletion residues 170–181 |
| Transfected construct (*Homo-sapiens*) | pCS2-human IGF1R WT-MYC | *Li et al., 2019*, Nature Communications | | |
| Transfected construct (*Homo-sapiens*) | pCS2-human IGF1R P673G4-MYC | This paper | | Four glycine insertion between P673 and K674 |
| Transfected construct (*Homo-sapiens*) | pCS2-human IGF1R Δ3C-MYC | This paper | | Deletion residues 669–572 |
| Transfected construct (*Homo-sapiens*) | pBabe-puro-IR WT-GFP | *Choi et al., 2016*, Cell | | |
| Transfected construct (*Homo-sapiens*) | pBabe-puro-IR 3CS-GFP | This paper | | Cysteine mutation (C682S, C683S, C685S) |

*Appendix 1 Continued on next page*

*Appendix 1 Continued*

| Reagent type (species) or resource | Designation | Source or reference | Identifiers | Additional information |
|---|---|---|---|---|
| Transfected construct (*Homo-sapiens*) | pBabe-puro-IR Δ686–690-GFP | This paper | | Deletion residues 686–690 |
| Transfected construct (*Homo-sapiens*) | pBabe-puro-IGF1R WT-GFP | This paper | | Mature human IGF1R |
| Transfected construct (*Homo-sapiens*) | pBabe-puro-IGF1R P673G4 | This paper | | Four glycine insertion between P673 and K674 |
| Antibody | Anti-IR-pY1150/1151 (Rabbit monoclonal, 19H7) | Cell Signaling | #3024 | WB (1:1000) |
| Antibody | Anti-AKT (Mouse monoclonal, 40D4) | Cell Signaling | #2920 | WB (1:1000) |
| Antibody | Anti-pS473 AKT (Rabbit monoclonal, D9E) | Cell Signaling | #4060 | WB (1:1000) |
| Antibody | Anti-ERK1/2 (Mouse monoclonal, L34F12) | Cell Signaling | #4696 | WB (1:1000) |
| Antibody | Anti-pERK1/2 (Rabbit monoclonal, 197G2) | Cell Signaling | #4377 | WB (1:1000) |
| Antibody | Anti-Myc (Mouse monoclonal, 9E10) | Roche | #11667149001 | WB (1:1000) |
| Antibody | Anti-GFP (Rabbit polyclonal) | Homemade (*Xia et al., 2004*; *Tang et al., 2001*) | | IFA (1:500) |
| Antibody | Anti-rabbit immunoglobulin G (IgG) (H+L) Dylight 800 conjugates (secondary antibody) | Cell Signaling | #5151 | WB (1:5000) |
| Antibody | Anti-mouse IgG (H+L) Dylight 680 conjugates (secondary antibody) | Cell Signaling | #5470 | WB (1:5000) |
| Recombinant DNA reagent | p-gag/pol | Addgene | #14887 | Retrovirus production |
| Recombinant DNA reagent | pCMV-VSV-G | Addgene | #8454 | Retrovirus production |
| Sequence-based reagent | PCR primers site-directed mutagenesis for human IR 3CS | This paper | | **F**:tcctctCCAAAGACAGACTCTCAGATCCTG **R**:ggaggaTTCGCCGGCCGAATCCTC |
| Sequence-based reagent | PCR primers site-directed mutagenesis for human IR Δ686–690 | This paper | | **F**:CAGATCCTGAAGGAGCTGG **R**:ACAGGAGCAGCATTCGCC |
| Sequence-based reagent | PCR primers site-directed mutagenesis for human IR Δ170–181 | This paper | | **F**:TGTTGGACTCATAGTCACTG **R**:GCAGTTGGTCTTGCCCTT |
| Sequence-based reagent | PCR primers site-directed mutagenesis for human IGF1R P673G4 | This paper | | **F**:ggaggtAAAACTGAAGCCGAGAAGCAGGCC **R**:acctccGGGGCAGGCGCAGCAAGG |
| Sequence-based reagent | PCR primers site-directed mutagenesis for human IGF1R Δ3C | This paper | | **F**:CCCAAAACTGAAGCCGAGAAG **R**:AGGCCCTTTCTCCCCACC |

*Appendix 1 Continued on next page*

*Appendix 1 Continued*

| Reagent type (species) or resource | Designation | Source or reference | Identifiers | Additional information |
|---|---|---|---|---|
| Sequence-based reagent | PCR primers site-directed mutagenesis for mouse IGF1R P673G4 | This paper | | **F**:CTGCGCTTGCCCTGGCGGAGGAGG CAAAACTGAAGCTGAGAAGCAGG **R**:GCTTCAGTTTTGCCTCCT CCGCCAGGGCAAGCGCAGCAT |
| Sequence-based reagent | PCR primers site-directed mutagenesis for mouse IR-3CS | This paper | | **F**:TCATCTCCTAAGACTGACTCTCAGATCC **R**:GGATGACTCACTGGCCGAGTCGTC |
| Peptide, recombinant protein | Human Insulin | Sigma | I2643 | |
| Peptide, recombinant protein | Human IGF1 | PeproTech | 100–11 | |
| Commercial assay or kit | Micro BCA Protein Assay Kit | Thermo Scientific | 23235 | |
| Commercial assay or kit | Alexa Fluor 488 | Thermo Scientific | A10235 | |
| Commercial assay or kit | Q5 site directed mutagenesis kit | NEB | E0554S | |
| Commercial assay or kit | Gibson Assembly Master Mix | NEB | E2166L | |
| Chemical compound, drug | cOmplete Protease Inhibitor Cocktail | Roche | 05056489001 | |
| Chemical compound, drug | PhosSTOP | Roche | 4906837001 | |
| Chemical compound, drug | BMS536924 | Tocris | 4774 | |
| Chemical compound, drug | Cellfectin | Invitrogen | 10362100 | |
| Chemical compound, drug | Lipofectamine 2000 | Invitrogen | 11668019 | |
| Software, algorithm | Prism 9.0d | GraphPad | N/A | Statistics |
| Other | Pierce Anti-c-Myc Magnetic Beads | Thermo Scientific | 88843 | In vitro insulin binding assay |
| Other | ProLong Gold Antifade reagent with DAPI | Invitrogen | P36935 | Immunofluorescence assay for IR and IGF1R trafficking |

