## [Editor Report]

This paper investigated the structural changes in the insulin and IGF1 receptors caused by ligand binding using cryo-electron microscopy and found that the disulfide-linked C-terminal segment of α-chains of these receptors are important for receptor activation. This is an important manuscript that addresses a research question of interest to the fields of metabolism, cancer, growth and aging. It provides convincing mechanistic insights into the roles of the disulfide linked C-terminal segment of the α-chains of the IR and IGF1R and their activation.

---

## [Decision Letter]

**Decision letter after peer review:**

Thank you for submitting your article "Molecular basis for the role of disulfide-linked αCTs in the activation of insulin-like growth factor 1 receptor and insulin receptor" for consideration by *eLife*. Your article has been reviewed by 3 peer reviewers, and the evaluation has been overseen by a Reviewing Editor and Mone Zaidi as the Senior Editor. The following individuals involved in review of your submission have agreed to reveal their identity: Pierre De Meyts (Reviewer #1); Briony E Forbes (Reviewer #2); Stevan R Hubbard (Reviewer #3).

Essential revisions:

1. The main concern from all reviewers related to how much the study truly advances our understanding of the mechanisms of activation of the IR and IGF1R, and why they differ.

2. There were some inconsistent data that is unexplained, such as why the IR-3CS has increased activation of pY1150/Y1151, but poor downstream signaling.

3. Some conclusions reported by the authors, including that the apparent differences in the conformational changes of the IR and IGF1R upon ligand binding is due to the lack of site 2 on the IGF-1R; however, the data in the manuscript did not support this conclusion.

*Reviewer #1 (Recommendations for the authors):*

This interesting article brings new and valuable data and concepts, based on cryoEM, regarding the fine details of ligand binding, negative cooperativity and activation of the insulin and IGF1 receptors (IR and IGF1Rs), a field in a dynamic state of evolution right now. The authors use an astute strategy to evaluate the role of the α-CT motives of the disordered disulfide-linked insert domains in the FnIII-2 modules of the receptors.

*Reviewer #2 (Recommendations for the authors):*

What are the IC50s or Kd derived from the binding data in Figure 5C? The affinity expected for WT IR is subnanomolar but by eye the IC50 for WT looks like about 10 nanomolar. Please include the IC50s.

The use of "strong" and "weak" to describe differences in negative cooperativity is a bit unclear. I wonder if this could be defined in more molecular terms at the first instance you mention it?

It would be good to refer to the biochemical data reported in Kiselyov et al. that show the differences in potency of negative cooperativity between the IGF-1R and IR when discussing the mechanisms of negative cooperativity.

I understood negative cooperativity to be characterised or identified in dissociation experiments where the dissociation of labelled insulin or IGF-I is accelerated by the addition of unlabelled ligand. The most common explanation of this is that binding of a second ligand to a different site promotes dissociation of the radioligand at the original site. I understand that the asymmetric conformation inhibits binding of a second IGF-I to site 1 but do you think that another IGF-I transiently interacts with a second site to accelerate the dissociation of the first? I think there needs to be a clearer explanation.

*Reviewer #3 (Recommendations for the authors):*

Overall, this study provides useful mechanistic information on the roles of the disulfide-linked alphaCT regions of IGF1R and IR in negative cooperativity and receptor activation.

What is still missing, though, in my opinion, is a molecular explanation as to why these two highly related receptors are activated in such distinct ways, with IGF1R activated by a single IGF1 molecule, in which the receptor adopts the asymmetric conformation, and with IR activated by four insulin molecules, in which the receptor adopts the symmetric T conformation. Clearly, the fact that the IR has a second pair of ligand binding sites is part of the explanation, but are there differences in the alphaCT regions of the two receptors that also contribute to the formation of the (disparate) active states and to the strength of ligand-binding negative cooperativity? Experiments (e.g., chimeras) that address this issue would increase the impact of this work. Otherwise, the study could be deemed an incremental advancement in our understanding of how these important receptors operate.

---

## [Author Response]

Essential revisions:1. The main concern from all reviewers related to how much the study truly advances our understanding of the mechanisms of activation of the IR and IGF1R, and why they differ.

The sequences of αCTs are not highly conserved between IR and IGF1R. In this work, we demonstrate that disulfide-linked αCTs play distinct roles in the activation of IR and IGF1R. In the insulin-mediated IR activation, the disulfide-linked αCTs promote the formation of T-shaped symmetric IR dimer at saturated insulin concentrations; whereas, in the IGF1-mediated IGF1R activation, the disulfide-linked αCTs are the source of negative cooperativity in the binding of IGF1 to IGF1R, and they also stabilize an active, single IGF1-bound conformation of IGF1R. Together, we propose that αCTs play a critical role in the distinct activation mechanism of IR and IGF1R. Therefore, we believe our work advance the understanding of the mechanism of activation of the IR and IGF1R.

In addition, it is important to note that IR and IGF1R share a high degree of structural similarity, but sequences in their extracellular domains are less conserved. For example, a β hairpin motif containing residues 163-174 of the L1 domain of IGF1R stabilizes the Γ-shaped asymmetric conformation by contacting the FnIII-2 domain of the same protomer. Such intra-protomer interaction is important for one IGF1-induced IGF1R activation, as the deletion of the entire L1 β hairpin (∆163-174) significantly reduced the IGF1-dependent activation of IGF1R (PMID: 31594955). The key residues for the L1-FnIII-2 interaction, such as K164 and Y173, are not conserved in IR. Contrary to IGF1R, our new results showed that deletion of the L1 β hairpin of IR (∆170-181, corresponding to IGF1R residence 163-174) did not affect insulin-dependent IR activation (Figure. 6-supplement 1). This suggests that not only the αCT regions of the receptors, but also other extracellular domains contribute to the distinct active conformations between IR and IGF1R. We have discussed this issue in the revised manuscript.

2. There were some inconsistent data that is unexplained, such as why the IR-3CS has increased activation of pY1150/Y1151, but poor downstream signaling.

Asymmetric IR-3CS/insulin, asymmetric IR/insulin and symmetric IR/insulin have similar distances between their membrane-proximal regions (approximately 30 – 35 Å). This indicates that the distances between the membrane-proximal regions within these complexes are all short enough to allow the intracellular kinase to undergo efficient autophosphorylation, in contrast to IGF1R. Nevertheless, our cellular functional assays showed that the IR-3CS has higher levels of autophosphorylation, but lower levels of downstream signaling activity and a defect in endocytosis. Although the distances between the membrane-proximal regions are similar, the relative positions and orientations between the two membrane proximal regions are significantly different between asymmetric IR-3CS and symmetric IR. Given the fact that the FnIII-3 domain is connected to the transmembrane domain by a short linker containing four residues, we speculate that the structural differences in the extracellular domains may lead to both differential dimeric assembly of transmembrane and intracellular domains, as well as the stable interaction between the intracellular IR domains and downstream adaptors and effectors. This could in part explain why IR-3CS can still undergo robust autophosphorylation but its downstream signaling becomes defective. Similar hypothesis has been proposed in the EGF and TGF-α induced activation of EGFR (PMID: 34846302). The endocytosis defects of IR-3CS might be the result of reduced IR signaling, but it is tempting to speculate that less endocytosis of IR-3CS may cause defective downstream signaling. The structure of transmembrane and intracellular domains in the context of the entire full-length/insulin complex needs to be further investigated. We have included new analysis (Figure 4-supplement 2) and expanded the discussion.

3. Some conclusions reported by the authors, including that the apparent differences in the conformational changes of the IR and IGF1R upon ligand binding is due to the lack of site 2 on the IGF-1R; however, the data in the manuscript did not support this conclusion.

Gauguin L. et al. demonstrated that alanine mutagenesis in IGF1, including E9A, D12A, F16A, D53A, L54A, and E58A, markedly reduced IGF1R-binding affinity. With the exception of IGF1 E9 (Site-1b), none of IGF1 D12, F16, D53, L54, and E58 are involved in the binding to site-1, suggesting that IGF1 has another binding site on IGF1R. Despite saturated IGF1 levels, however, our previous and current structural studies did not reveal the secondary IGF1 binding site. We speculate that IGF1 may bind to site-2 transiently during activation. In addition, as we noted above, not only the αCT regions of the receptors, but also other extracellular domains contribute to distinct active conformations and negative cooperativity during receptor activation. We have revised the manuscript thoroughly and expanded the discussion.

Reviewer #2 (Recommendations for the authors):What are the IC50s or Kd derived from the binding data in Figure 5C? The affinity expected for WT IR is subnanomolar but by eye the IC50 for WT looks like about 10 nanomolar. Please include the IC50s.

It is important to note that due to the detection limitation, we used 0.5 nM of Alexa 488 labelled insulin for the competition assay, which differs greatly from the Kiselyov et al. (the hot ligand was used 7 pM). In our condition, IC50 for WT is 4.260±0.9109 nM, (Mean ± SD), IC50 for IR-3CS is 3.306±0.4619 nM, and IC50 for IR-∆686-690 is 3.231±0.06734 nM. We have included the IC50 values in the revised manuscript.

The use of "strong" and "weak" to describe differences in negative cooperativity is a bit unclear. I wonder if this could be defined in more molecular terms at the first instance you mention it?

Point accepted. The description has been revised throughout the manuscript.

It would be good to refer to the biochemical data reported in Kiselyov et al. that show the differences in potency of negative cooperativity between the IGF-1R and IR when discussing the mechanisms of negative cooperativity.

We thank Dr. Forbes for this suggestion. We have cited the reference and expanded discussion.

I understood negative cooperativity to be characterised or identified in dissociation experiments where the dissociation of labelled insulin or IGF-I is accelerated by the addition of unlabelled ligand. The most common explanation of this is that binding of a second ligand to a different site promotes dissociation of the radioligand at the original site. I understand that the asymmetric conformation inhibits binding of a second IGF-I to site 1 but do you think that another IGF-I transiently interacts with a second site to accelerate the dissociation of the first? I think there needs to be a clearer explanation.

We thank Dr. Forbes for this great suggestion. Although our previous and current structural analysis did not reveal the second IGF1 binding sites, it is possible that transient IGF1 binding to the receptor (*e.g.* another site-1 or unknown site-2) accelerates the dissociation of the first IGF1. We have included this possibility in the revised manuscript.

Reviewer #3 (Recommendations for the authors):Overall, this study provides useful mechanistic information on the roles of the disulfide-linked alphaCT regions of IGF1R and IR in negative cooperativity and receptor activation.What is still missing, though, in my opinion, is a molecular explanation as to why these two highly related receptors are activated in such distinct ways, with IGF1R activated by a single IGF1 molecule, in which the receptor adopts the asymmetric conformation, and with IR activated by four insulin molecules, in which the receptor adopts the symmetric T conformation. Clearly, the fact that the IR has a second pair of ligand binding sites is part of the explanation, but are there differences in the alphaCT regions of the two receptors that also contribute to the formation of the (disparate) active states and to the strength of ligand-binding negative cooperativity? Experiments (e.g., chimeras) that address this issue would increase the impact of this work. Otherwise, the study could be deemed an incremental advancement in our understanding of how these important receptors operate.

We thank Dr. Stevan Hubbard for this suggestion. IR and IGF1R share a high degree of structural similarity, but sequences in their extracellular domains are less conserved. For example, a β hairpin motif containing residues 163-174 of the L1 domain of IGF1R stabilizes the Γ-shaped asymmetric conformation by contacting the FnIII-2 domain of the same protomer. Such intra-protomer interaction is important for one IGF1 induced IGF1R activation, as the deletion of the entire L1 β hairpin (∆163-174) significantly reduced the IGF1-dependent activation of IGF1R (PMID: 31594955). The key residues for the L1-FnIII-2 interaction, such as K164 and Y173, are not conserved in IR. Contrary to IGF1R, our new results showed that deletion of the L1 β hairpin of IR (∆170-181, corresponding to IGF1R residence 163-174) did not affect insulin-dependent IR activation (Figure. 6-supplement 1). This suggests that not only the αCT regions of the receptors, but also other extracellular domains contribute to the distinct active conformations between IR and IGF1R. In accordance with Dr. Hubbard's suggestion, we attempted to design experiments using chimeras (*e.g.* IR with the αCT of IGF1R or IGF1R with the αCT of IR; (2) IR/IGF1R chimera). The results would be difficult to interpret, however, since swapping the αCTs between IR and IGF1R will also change the affinity of ligand binding. The percentage of the receptors that could form the IR/IGF1R chimera is also difficult to predict, which would hinder the interpretation of the results from the cell-based experiment. Hope Dr. Hubbard would agree with us about the limitations of these experiments. In the revised manuscript, we have included new results (Figure. 6-supplement 1) and expanded the discussion.